# Advancing Bayesian Optimization via Learning Correlated Latent Space

**Seunghun Lee,** * **Jaewon Chu,** * **Sihyeon Kim,** * **Juyeon Ko,** **Hyunwoo J. Kim** †
Computer Science & Engineering
Korea University
{llsshh319, allonsy07, sh_bs15, juyon98, hyunwoojkim}@korea.ac.kr

## Abstract

Bayesian optimization is a powerful method for optimizing black-box functions with limited function evaluations. Recent works have shown that optimization in a latent space through deep generative models such as variational autoencoders leads to effective and efficient Bayesian optimization for structured or discrete data. However, as the optimization does not take place in the input space, it leads to an inherent gap that results in potentially suboptimal solutions. To alleviate the discrepancy, we propose **Cor**related latent space **B**ayesian **O**ptimization (**CoBO**), which focuses on learning correlated latent spaces characterized by a strong correlation between the distances in the latent space and the distances within the objective function. Specifically, our method introduces Lipschitz regularization, loss weighting, and trust region recoordination to minimize the inherent gap around the promising areas. We demonstrate the effectiveness of our approach on several optimization tasks in discrete data, such as molecule design and arithmetic expression fitting, and achieve high performance within a small budget.

## 1 Introduction

Bayesian optimization (BO) is a standard method for a wide range of science and engineering problems such as chemical design [1–4], reinforcement learning [5], and hyperparameter tuning [6]. Relying on a surrogate model typically modeled with a Gaussian process (GP), BO estimates the computationally expensive black-box objective function to solve the problem with a minimum number of function evaluations [7]. While it is known as a powerful method on continuous domains [8, 9], applying BO is often obstructed by structured or discrete data, as the objective values are in a complex combinatorial space [10]. This challenge has motivated recent interest in latent space Bayesian optimization (LBO) methods [11–14], which aim to find solutions in low-dimensional, continuous embeddings of the input data. By adopting deep generative models such as variational autoencoders (VAEs) [15] to map the input space to the latent space, LBO has successfully addressed the difficulties of such optimization problems.

However, the fact that optimization is not directly conducted in the input space gives rise to inherent gaps, which may lead to failures in the optimization process. First, we can think of the gap between the input space and the latent space. Seminal works have been developed to address this gap, with generative models such as $\beta$-VAE [16] emphasizing the importance of controlling loss weights, while WGAN [17] introduces improved regularization. Both aim to learn the improved latent space that better aligns with the input data distribution. Second, considering the BO problems, an additional gap emerges between the proximity of solutions in the latent space and the similarity of their

---

*equal contributions
†Corresponding author

37th Conference on Neural Information Processing Systems (NeurIPS 2023).

black-box objective function values. This is observed in many prior works [18, 19] that learn a latent space by minimizing only reconstruction errors without considering the surrogate model. This often leads to suboptimal optimization results. A recent study [10] has highlighted the significance of joint training between VAE and the surrogate model, yet it only implicitly encourages the latent space to align with the surrogate model. This limitation is observed in Figure 5a, where the latent space's landscape, with respect to objective values, remains highly non-smooth with joint training.

To this end, we propose our method **Cor**related latent space **B**ayesian **O**ptimization (**CoBO**) to address the inherent gaps in LBO. First, we aim to minimize the gap between the latent space and the objective function by increasing the correlation between the distance of latent vectors and the differences in their objective values. By calculating the lower bound of the correlation, we introduce two regularizations and demonstrate their effectiveness in enhancing the correlation. Especially, one of these regularizations, called the Lipschitz regularization, encourages a smoother latent space, allowing for a more efficient optimization process (see Figure 5b). Moreover, we suggest loss weighting with respect to objective values of each input data point to particularly minimize the gap between the input space and the latent space around promising areas of high objective values. Finally, we propose the concept of trust region recoordination to adjust the search space in line with the updated latent space. We experimentally validate our method with qualitative and quantitative analyses on nine tasks using three benchmark datasets on molecule design and arithmetic expression fitting.

To summarize, our contributions are as follows:

- We propose Correlated latent space Bayesian Optimization (CoBO) to bridge the inherent gap in latent Bayesian optimization.

- We introduce two regularizations to align the latent space with the black-box objective function based on increasing the lower bound of the correlation between the Euclidean distance of latent vectors and the distance of their corresponding objective values.

- We present a loss weighting scheme based on the objective values of input points, aiming to close the gap between the input space and the latent space focused on promising areas.

- We demonstrate extensive experimental results and analyses on nine tasks using three benchmark datasets on molecule design and arithmetic expression fitting and achieve state-of-the-art in all nine tasks.

## 2 Methods

In this section, we describe the main contributions of our method. Section 2.1 introduces several preliminaries on Bayesian optimization. In Section 2.2, we propose two regularizations to align the latent space with the black-box objective function. In Section 2.3, we describe our loss weighting scheme with the objective values. Lastly, in Section 2.4, we explain the overall architecture of our method.

### 2.1 Preliminaries

**Bayesian optimization.** Bayesian optimization (BO) [6, 8, 9] is a classical, sample-efficient optimization method that aims to solve the problem

$$\mathbf{x}^* = \underset{\mathbf{x} \in \mathcal{X}}{\arg\max} f(\mathbf{x}), \tag{1}$$

where $\mathcal{X}$ is a feasible set and $f$ is a black-box objective function. Since the function evaluation is assumed to be expensive, BO constructs a probabilistic model of the black-box objective function. There are two main components of BO, first is a surrogate model $g$ that provides posterior probability distribution over $f(\mathbf{x})$ conditioned on observed dataset $\mathcal{D} = \{(\mathbf{x}_i, y_i)\}_{i=1}^{n}$ based prior over objective function. Second is an acquisition function $\alpha$ for deciding the most promising next query point $\mathbf{x}_{i+1}$ based on the posterior distribution over $f(\mathbf{x})$. BO is a well-established method, however, applying BO to high-dimensional data can be challenging due to the exponential growth of the search space. To alleviate the challenge, recent approaches [10, 20] restrict the search space to a hyper-rectangular trust region centered on the current optimal input data point. In this paper, we adopt this trust-region-based BO for handling high-dimensional search space.

**Latent space Bayesian optimization.** BO over structured or discrete input space $\mathcal{X}$ is particularly challenging, as a search space over the objective function becomes a large combinatorial one. In an effort to reduce a large combinatorial search space to continuous space, latent space Bayesian optimization (LBO) [10, 13, 14, 19, 21, 18] suggests BO over continuous latent space $\mathcal{Z}$. A pretrained VAE $= \{q_\phi, p_\theta\}$ [15] is commonly used as the mapping function, where the latent space is learned to follow the prior distribution (Gaussian distribution). Given a pretrained VAE, an encoder $q_\phi : \mathcal{X} \mapsto \mathcal{Z}$ maps the input $\mathbf{x}_i$ to the latent vector $\mathbf{z}_i$ and the surrogate model $g$ takes $\mathbf{z}_i$ as the input. After the acquisition function $\alpha$ suggests the next latent query point $\mathbf{z}_{i+1}$, a decoder $p_\theta : \mathcal{Z} \mapsto \mathcal{X}$ reconstructs $\mathbf{z}_{i+1}$ to $\mathbf{x}_{i+1}$, so that it can be evaluated by the black-box objective function, *i.e.*, $f(\mathbf{x}_{i+1})$.

LBO is a promising optimization approach for discrete or structured inputs, yet, there are two main gaps to be considered. Firstly, there is a gap between the input space and the latent space. Our focus is on narrowing the gap, especially within the promising areas, *i.e.*, samples with high objective function values. Secondly, a gap exists between the proximity of solutions within the latent space and the similarity of their corresponding objective function values. This arises because the objective value originates from a black-box function in the discrete input space $\mathcal{X}$, distinct from the latent space $\mathcal{Z}$ where our surrogate model $g$ is learned. In the previous work, [10] has suggested closing the gap by jointly optimizing VAE and the surrogate GP model to align the latent space with current top-$k$ samples.

Here, we propose CoBO that explicitly addresses those two gaps by training a latent space with Lipschitz regularization, which increases the correlation between the distance of latent samples and the distance of objective values, and loss weighting with objective values to focus on relatively important search space.

## 2.2 Aligning the latent space with the objective function

Our primary goal is to align the latent space $\mathcal{Z}$ with the black-box objective function $f$, which can be achieved by increasing the correlation between the Euclidean distance of latent vectors and the differences in their corresponding objective values, *i.e.*, $\text{Corr}(||\mathbf{z}_1 - \mathbf{z}_2||_2, |y_1 - y_2|)$. Assuming the objective function $f$ is an $L$-Lipschitz continuous function, we can establish a lower bound of $\text{Corr}(||\mathbf{z}_1 - \mathbf{z}_2||_2, |y_1 - y_2|)$. In general, if the function $f$ is $L$-Lipschitz continuous, it is defined as

$$\forall \mathbf{z}_1, \mathbf{z}_2 \in \mathbb{R}^n, \quad d_Y(f(\mathbf{z}_1), f(\mathbf{z}_2)) \leq L d_Z(\mathbf{z}_1, \mathbf{z}_2), \tag{2}$$

where $d_Z$ and $d_Y$ are distance metrics in their respective spaces for $\mathbf{z}$ and $y$. Then, the lower bound of $\text{Corr}(||\mathbf{z}_1 - \mathbf{z}_2||_2, |y_1 - y_2|)$ can be obtained as Theorem 1.

**Theorem 1.** *Let $D_Z = d_Z(Z_1, Z_2)$ and $D_Y = d_Y(f(Z_1), f(Z_2))$ be random variables where $Z_1, Z_2$ are i.i.d. random variables, $f$ is an $L$-Lipschitz continuous function, and $d_Z, d_Y$ are distance functions. Then, the correlation between $D_Z$ and $D_Y$ is lower bounded as*

$$D_Y \leq L D_Z \Rightarrow Corr_{D_Z, D_Y} \geq \frac{\frac{1}{L}(\sigma_{D_Y}^2 + \mu_{D_Y}^2) - L\mu_{D_Z}^2}{\sqrt{\sigma_{D_Z}^2 \sigma_{D_Y}^2}},$$

*where $\mu_{D_Z}$, $\sigma_{D_Z}^2$, $\mu_{D_Y}$, and $\sigma_{D_Y}^2$ are the mean and variance of $D_Z$ and $D_Y$ respectively.*

Theorem 1 implies that under the assumption of the $L$-Lipschitz continuity of $f$, we can increase the lower bound of $\text{Corr}(||\mathbf{z}_1 - \mathbf{z}_2||_2, |y_1 - y_2|)$ by reducing Lipschitz constant $L$ while $\mu_{D_Z}^2$, $\sigma_{D_Z}^2$, $\mu_{D_Y}^2$, and $\sigma_{D_Y}^2$ remain as constants. Based on Theorem 1, we propose two regularizations. The first one is Lipschitz regularization, which encourages a *smooth* latent space $\mathcal{Z}$ w.r.t. the objective function $f$.

$$\mathcal{L}_{\text{Lip}} = \sum_{i,j \leq N} \max\left(0, \frac{|y_i - y_j|}{||\mathbf{z}_i - \mathbf{z}_j||_2} - L\right), \tag{3}$$

where $N$ is the number of training data points. Here, we set the Lipschitz constant $L$ as the median of all possible gradients of slopes. By penalizing slopes with larger gradients than an adaptively adjusted $L$, we encourage a decrease in $L$ itself, leading to learning a correlated latent space.

Next, it is beneficial to keep $\mu_{D_Z}^2$, $\sigma_{D_Z}^2$, $\mu_{D_Y}^2$ and $\sigma_{D_Y}^2$ as constants. Given that $\mu_{D_Y}^2$ and $\sigma_{D_Y}^2$ are function-specific constants, where the black-box function is unchanged throughout the optimization,

we can treat them as fixed values. Then, we want to constrain $\mu^2_{D_z}$ as a constant with the second regularization $\mathcal{L}_z$, by penalizing the average distance between the latent vectors $\mathbf{z}$ to be a constant $c$:

$$\mathcal{L}_{\mathrm{z}} = \left| \left( \frac{1}{N^2} \sum_{i,j \leq N} ||\mathbf{z}_i - \mathbf{z}_j||_2 \right) - c \right|. \tag{4}$$

We set $c$ as the expected Euclidean norm between two standard normal distributions, which is the prior of the variational autoencoder. That is the mean of the noncentral chi distribution [22] which is sum of squared independent normal random variables:

$$c = \mathbb{E}\left[ \sqrt{\Sigma_i^n (U_i - V_i)^2} \right] = \mathbb{E}\left[ C \right] = \frac{2\Gamma(\frac{k+1}{2})}{\Gamma(\frac{k}{2})}, U_i, V_i \sim \mathcal{N}(0,1), C \sim NC_{\chi_k}, \tag{5}$$

$$\sqrt{\Sigma_i^n (U_i - V_i)^2} = \sqrt{\Sigma_i^n W_i^2} = C, \ W_i \sim \mathcal{N}\left(0, \sqrt{2}^2\right), \tag{6}$$

where $\Gamma(\cdot)$ denotes the gamma function, C denotes the random variable with noncentral chi distribution $NC_{\chi_k}$ and $k$ denotes the degrees of freedom which is the same value as dimension $n$ of the latent vector. Then $c$ is dependent only on the dimension of the latent vector, $\mathbf{z} \in \mathbb{R}^n$. For $\sigma^2_{D_z}$, preliminary observations indicate that it stays in a reasonable range as long as $\mathcal{L}_{\mathrm{Lip}}$ is not overly penalized. Thus, we safely conclude that there is no need to explicitly constrain $\sigma^2_{D_z}$. We refer to the supplement for further analysis and the proof of Theorem 1.

## 2.3 Loss weighting with objective values

Our focus now shifts to addressing the gap between the input space $\mathcal{X}$ and the latent space $\mathcal{Z}$ for LBO. Especially, we aim to minimize the gap in promising areas that offer better optimization opportunities, *i.e.*, significant points with high objective values. To achieve this, we prioritize input data points based on their respective objective values by weighting the reconstruction loss term. Following [23], we utilize the cumulative density function of the Gaussian distribution for the weighting. Specifically, the weighting function w.r.t. objective value $y$ is:

$$\lambda(y) = P(Y > y_q), \tag{7}$$

with $Y \sim \mathcal{N}(y, \sigma^2)$, where $y_q$ represents a specific quantile of the distribution of $Y$, and hyperparameter $\sigma$ denotes the standard deviation of $Y$. The weighted reconstruction loss is as follows:

$$\mathcal{L}_{\mathrm{recon\_W}} = \lambda(y)\mathcal{L}_{\mathrm{recon}} = -\lambda(y)\mathbb{E}_{\mathbf{z} \sim q_\phi(\mathbf{z}|\mathbf{x})}[\log p_\theta(\mathbf{x}|\mathbf{z})]. \tag{8}$$

Moreover, we also apply the weighting scheme to the Lipschitz regularization term to promote a smoother latent space when the objective value is higher. The weighted Lipschitz regularization is defined with the geometric mean of the weights of two input data points:

$$\mathcal{L}_{\mathrm{Lip\_W}} = \sum_{i,j \leq N} \sqrt{\lambda(y_i)\lambda(y_j)} \max\left( 0, \frac{|y_i - y_j|}{||\mathbf{z}_i - \mathbf{z}_j||_2} - L \right). \tag{9}$$

## 2.4 Overall architecture of CoBO

In this section, we explain the overall architecture of CoBO. We first introduce the training schema of latent space in CoBO to encourage a high correlation between the distance in the latent space and the distance within the objective function. Next, we describe updating strategy of the surrogate model for modeling the black-box function and further present a generating procedure of the next candidate inputs for the black-box objective function through the acquisition function in the trust region. The overall procedure of our CoBO is in Algorithm 1.

**Learning the latent space.** Our method learns the latent space by optimizing the encoder $q_\phi$ and decoder $p_\theta$ of the pretrained VAE and updating the surrogate model in the latent space with our final loss:

$$\mathcal{L}_{\mathrm{CoBO}} = \mathcal{L}_{\mathrm{Lip\_W}} + \mathcal{L}_{\mathrm{z}} + \mathcal{L}_{\mathrm{recon\_W}} + \mathcal{L}_{\mathrm{KL}} + \mathcal{L}_{\mathrm{surr}}, \tag{10}$$

$$\mathcal{L}_{\mathrm{KL}} = \mathrm{KL}(q_\phi(\mathbf{z}|\mathbf{x})||p_\theta(\mathbf{z})), \tag{11}$$

where $\mathcal{L}_{\text{Lip\_W}}$ and $\mathcal{L}_{\text{z}}$ is the regularization term in the latent space at Section 2.2 and 2.3, $\mathcal{L}_{\text{recon\_W}}$ is the weighted reconstruction loss term, $\mathcal{L}_{\text{KL}}$ is the KL divergence between the latent space distribution and the prior, and $\mathcal{L}_{\text{surr}}$ is the loss for optimizing the surrogate model. We adopt the joint training scheme, training the surrogate model and the encoder $q_\phi$ of the VAE model jointly [10]. Under computational considerations, we retrain a latent space after $N_{\text{fail}}$ accumulated failure of updating the optimal objective value.

**Updating the surrogate model.** After jointly optimizing the latent space and the surrogate model, we freeze the parameter of VAE and train our surrogate model. Note that this update is executed only after consecutive failures of updating optimal value, we also train the surrogate model in every iteration. As exact Gaussian process (GP), *i.e.*, $f(\mathbf{x}) \sim \mathcal{GP}(m(\mathbf{x}), k(\mathbf{x}, \mathbf{x}'))$, where $m$ is a mean function and $k$ is a covariance function, is infeasible to handle large datasets due to cubic computational complexity $O(N^3)$ for $N$ data points, we employ sparse GP [24] as a surrogate model which is computationally efficient via inducing point method. To alleviate cubic complexity, sparse GP approximates black-box function with $M \ll N$ pseudo-training samples called 'inducing points' that reduce complexity to $O(MN^2)$. We select the most widely used RBF kernel as a sparse GP kernel function. Finally, we adopted deep kernel learning (DKL) [25] in conjunction with sparse GP for our final surrogate model.

**Generating candidates through acquisition function.** Candidate samples for the acquisition function are determined by random points in a trust region centered on the current optimal value. We take a simple and powerful method, Thompson sampling as an acquisition function within the context of Bayesian optimization. The surrogate model acts as the prior

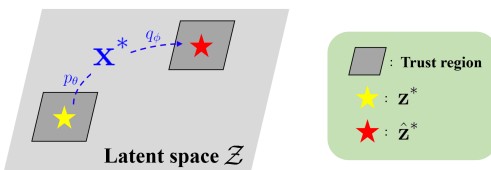

Figure 1: **Trust region recoordination.**

belief about our objective function. Thompson sampling uses this model to draw samples, leveraging its uncertainty to decide the next point to evaluate. In detail, we first select candidate samples in the trust region and score each candidate based on the posterior of the surrogate model to get the most promising values. Also, we recoordinate the center of the trust region to $\hat{\mathbf{z}}^*$, which is obtained by passing the current optimal latent vector $\mathbf{z}^*$ into updated VAE, *i.e.*, $q_\phi(p_\theta(\mathbf{z}^*))$, as shown in Figure 1. We empirically showed that trust region recoordination helps to find a better solution within a limited budget (see Table 2). Following [20], the base side length of the trust region is halved after consecutive failures of updating optimal objective value and doubled after consecutive successes.

## 3 Experiments

In this section, we demonstrate the effectiveness and efficiency of CoBO through various optimization benchmark tasks. We first introduce tasks and baselines. Then, in Section 3.1, we present evaluations of our method and the baselines for each task. In Section 3.2, we conduct an ablation study on the components of our approach. Finally, in Section 3.3, we provide qualitative analyses on the effects of our suggested regularizations and the necessity of z regularization.

**Tasks.** We evaluate CoBO to nine tasks on a discrete space in three different Bayesian optimization benchmarks, which consist of arithmetic expression fitting tasks, molecule design tasks named dopamine receptor D3 (DRD3) in Therapeutics Data Commons (TDC) [26] and Guacamol benchmarks [27]. The arithmetic expression task is generating polynomial expressions that are close to specific target expressions (*e.g.*, $1/3 + x + \sin(x \times x)$) [10, 11, 13, 14, 28], we set the number of initialization points $|D_0|$ to 40k, and max oracle calls to 500. In Guacamol benchmark, we select seven challenging tasks to achieve high objective value, Median molecules 2, Zaleplon MPO, Perindopril MPO, Osimertinib MPO, Ranolazine MPO, Aripiprazole similarity, and Valsartan SMART. The results for the last three tasks are in the supplement. The goal of each task is to find molecules that have the most required properties. For every task of Guacamol benchmark, we set the number of initialization points to 10k, and max oracle calls to 70k. DRD3 task in the TDC benchmark aims to find molecules with the largest docking scores to a target protein. In DRD3, the number of initialization points is set to 100, and the number of oracle calls is set to 3k. We use SELFIES VAE [10] in Chemical design, and Grammar VAE [11] in arithmetic expression.

**Algorithm 1** Correlated Bayesian Optimization (CoBO)

---

**Input:** Pretrained VAE encoder $q_\phi$, decoder $p_\theta$, black-box function $f$, surrogate model $g$, acquisition function $\alpha$, previously evaluated dataset $D_0 = \{(\mathbf{x}_i, y_i, \mathbf{z}_i)\}_{i=1}^n$, oracle budget $T$, latent update interval $N_{\text{fail}}$, batch size $N_b$, loss for surrogate model $\mathcal{L}_{\text{surr}}$, proposed loss for joint training $\mathcal{L}_{\text{CoBO}}$

1: $D \leftarrow D_0$
2: $n_{\text{fail}} \leftarrow 0$
3: **for** $t = 1, 2, ..., T$ **do**
4:     $D' \leftarrow D[-N_b :] \cup \text{top-}k(D)$
5:     **if** $n_{\text{fail}} \geq N_{\text{fail}}$ **then**
6:         $n_{\text{fail}} \leftarrow 0$
7:         Train $q_\phi, p_\theta, g$ with $\mathcal{L}_{\text{CoBO}}, D'$                 ▷ *Eq. 10*
8:         $Z \leftarrow \{q_\phi(\mathbf{x}_i) | (\mathbf{x}_i, y_i, \mathbf{z}_i) \in D'\}$
9:         $D \leftarrow D \cup \{(p_\theta(\mathbf{z}_i), f(p_\theta(\mathbf{z}_i)), \mathbf{z}_i) | \mathbf{z}_i \in Z\}$
10:     **end if**
11:     Train $g$ with $\mathcal{L}_{\text{surr}}, D'$ if $t \neq 1$ else $D_0$
12:     $(\mathbf{x}^*, y^*, \mathbf{z}^*) = \arg\max_{(\mathbf{x}, y, \mathbf{z}) \in D} y$
13:     $\hat{\mathbf{z}}^* \leftarrow q_\phi(p_\theta(\mathbf{z}^*))$                          ▷ *trust region recoordination*
14:     Get a candidate set $Z_{\text{cand}}$ with random points in the trust region around $\hat{\mathbf{z}}^*$
15:     $\mathbf{z}_{\text{next}} \leftarrow \text{argmax}_{\mathbf{z} \in Z_{\text{cand}}} \alpha(\mathbf{z})$
16:     **if** $f(p_\theta(\mathbf{z}_{\text{next}})) \leq y^*$ **then** $n_{\text{fail}} \leftarrow n_{\text{fail}} + 1$
17:     $D \leftarrow D \cup \{(p_\theta(\mathbf{z}_{\text{next}}), f(p_\theta(\mathbf{z}_{\text{next}})), \mathbf{z}_{\text{next}})\}$
18: **end for**
19: **return** $\mathbf{x}^*$

---

**Baselines.** We compare our methods with four BO baselines: LOL-BO [10], W-LBO [13], TuRBO [20] and LS-BO. LOL-BO proposed the joint loss to close the gap between the latent space and the discrete input space. Additionally, W-LBO weighted data points to emphasize important samples with regard to their objective values. To handle discrete and structured data, we leverage TuRBO in latent space, TuRBO-$L$ through encoder $q_\phi$ of pretrained frozen VAE and reconstruct it by decoder $p_\theta$ to be evaluated by the objective function. For more details about TuRBO-$L$, see [10]. In our LS-BO approach, we adopt the standard LBO methodology. Specifically, we use a VAE with pre-trained parameters that are frozen. For sampling, the candidates are sampled from a standard normal Gaussian distribution, $\mathcal{N}(0, I)$.

### 3.1 Results on diverse benchmark tasks

Figure 2, 3 represent the graphs that depict the number of oracle calls, *i.e.*, the number of the black-box function evaluations, and the corresponding mean and standard deviation of objective value. The maximum number of oracles is set to 70k with Gucamol benchmarks, 3k with the DRD3 benchmark, and 500 with the arithmetic expression task. As shown in Figure 2, our method outperformed all the baselines in all the tasks. Note that the goal of the arithmetic task in Figure 3a and the DRD3 task Figure 3b is minimizing the objective function, while others aim to maximize it. In the case of Figure 2a and Figure 2d, LS-BO and TuRBO-$L$ markedly failed to search for the one that has the best score. Especially in Figure 2a, LS-BO and TuRBO-$L$, which only search with the fixed latent space, failed to update the initial best point despite searching 70k molecules. It implies that the pretrained VAE cannot generate better molecules unless the latent space is updated.

### 3.2 Ablation study

Table 1: **An ablation of CoBO's components.**

| $\mathcal{L}_{\text{z}}$ | $\lambda(y)$ | $\mathcal{L}_{\text{Lip}}$ | Score |
|:---:|:---:|:---:|:---:|
| ✓ | ✓ | ✓ | 0.7921 |
| × | ✓ | ✓ | 0.7835 |
| × | × | ✓ | 0.7733 |
| × | × | × | 0.7504 |

Table 2: **An ablation on trust region recoordination.**

| Recoordination | Score |
|:---:|:---:|
| ✓ | 0.7921 |
| × | 0.6844 |

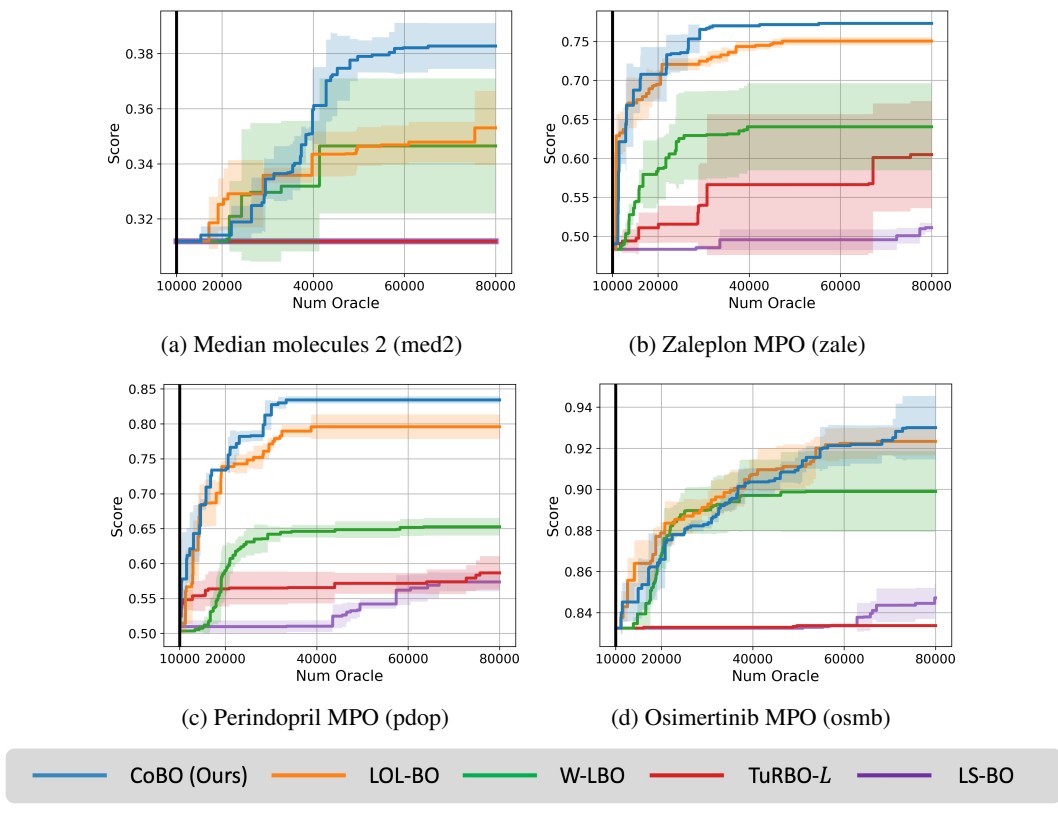

(a) Median molecules 2 (med2)  (b) Zaleplon MPO (zale)

(c) Perindopril MPO (pdop)  (d) Osimertinib MPO (osmb)

Figure 2: **Optimization results with four different tasks on the Guacamol benchmark.** The lines and range are the mean and standard deviation of three repetitions with the same parameters.

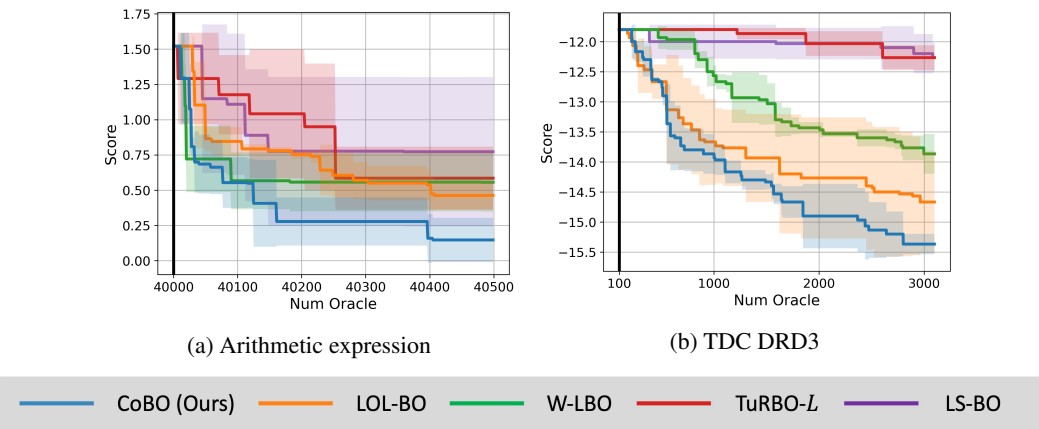

(a) Arithmetic expression  (b) TDC DRD3

Figure 3: **Optimization results with the arithmetic expression and TDC DRD3 benchmark.** The lines and range are the mean and standard deviation of three repetitions with the same parameters.

In this section, we evaluate the main components of our model to analyze their contribution. We employ Perindopril MPO (pdop) task for our experiment and note that all scores of ablation studies are recorded when the oracle number is 20k to compare each case in limited oracle numbers as the original BO intended.

We ablate the three main components of our CoBO and report the results in Table 1. As latent space regularization $\mathcal{L}_z$ comes from Lipschitz regularization and loss weighting schema aims to prioritize penalized input points, we conducted cascading experiments. Notably, we observed that performance decreased as the components were omitted, and the largest performance dropped (0.0229)

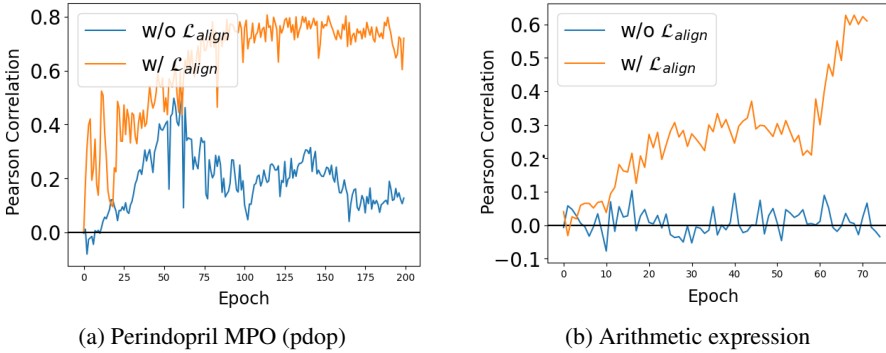

(a) Perindopril MPO (pdop)                    (b) Arithmetic expression

Figure 4: **Effects of $\mathcal{L}_{\mathbf{align}}$.** The plot depicts the Pearson correlation between the distance of the latent vectors and the distance of the objective values over the Bayesian optimization process. Each line represents training with a different loss for the latent space during the optimization process: one with $\mathcal{L}_{\text{align}}$ (orange) and the other without $\mathcal{L}_{\text{align}}$ (blue), where $\mathcal{L}_{\text{align}} = \mathcal{L}_{\text{Lip}} + \mathcal{L}_{\text{z}}$. We measure the correlation after every VAE update.

when Lipschitz regularization was not employed. Table 2 reports the effectiveness of trust region re-coordination. We observe that applying trust region recoordination makes a considerable difference within a small oracle budget, as the score increases 0.6844 to 0.7921 with recoordination.

### 3.3 Analysis on proposed regularizations

All the analyses were conducted on the Perindopril MPO (pdop) task from the Guacamol benchmark. We include an additional analysis with the arithmetic data. For convenience, we define the $\mathcal{L}_{\text{align}}$ as follows:

$$\mathcal{L}_{\text{align}} = \mathcal{L}_{\text{Lip}} + \mathcal{L}_{\text{z}}, \tag{12}$$

which is the loss that aligns the latent space with the black-box objective function.

**Effects of $\mathcal{L}_{\mathbf{align}}$ on correlation.** In Section 2.2, we theoretically prove that $\mathcal{L}_{\text{align}}$ increases the correlation between the distance in the latent space and the distance within the objective function. Here, we demonstrate our theorem with further quantitative analysis to show correlation changes during the Bayesian optimization process. Figure 4 shows Pearson correlation value between the latent space distance $\|\mathbf{z}_i - \mathbf{z}_j\|_2$ and the objective value distance $|y_i - y_j|$. The blue and orange lines indicate the models with and without our align loss $\mathcal{L}_{\text{align}}$ in Eq. 12, respectively. We measure the correlation with $10^3$ data point and every $10^6$ pair. The data is selected as the top $10^3$ points with the highest value among $10^3$ data, which are from training data for the VAE model and surrogate model. Over the training process, the Pearson correlation values with our align loss $\mathcal{L}_{\text{align}}$ were overall higher compared to the baseline. Moreover, $\mathcal{L}_{\text{align}}$ increases the Pearson correlation value high over $0.7$ in Figure 4a which is normally regarded as the high correlation value. This means align loss increases the correlation between the distance of the latent vectors and the distance of the objective values effectively, leading to narrowing the gap between the latent space and the objective function.

**Effects of $\mathcal{L}_{\mathbf{align}}$ on smoothness.** To validate that our model encourages the smooth latent space, we qualitatively analyze our model with $\mathcal{L}_{\text{align}}$ and without $\mathcal{L}_{\text{align}}$ by visualizing the landscape of the latent space after the training finishes. In Figure 5, we visualize the top-$k$ objective value and the corresponding latent vector in 2D space with the 2D scatter plot and the corresponding 3D plot for better understanding. We reduce the dimension of the latent vector to two dimensions using principal component analysis (PCA). The color means the normalized relative objective score value. The landscape of objective value according to the latent space with $\mathcal{L}_{\text{align}}$ (right) is smoother than the case without $\mathcal{L}_{\text{align}}$ (left). It demonstrates our $\mathcal{L}_{\text{align}}$ loss encourages smoothness of the latent space with respect to the objective function. Note that due to the inherent discreteness of the black-box objective function, it's expected to observe some spaces between the clusters in the 2D plot. Still, this does not detract from our purpose of effectively aligning latent vectors with respect to their discrete objective values.

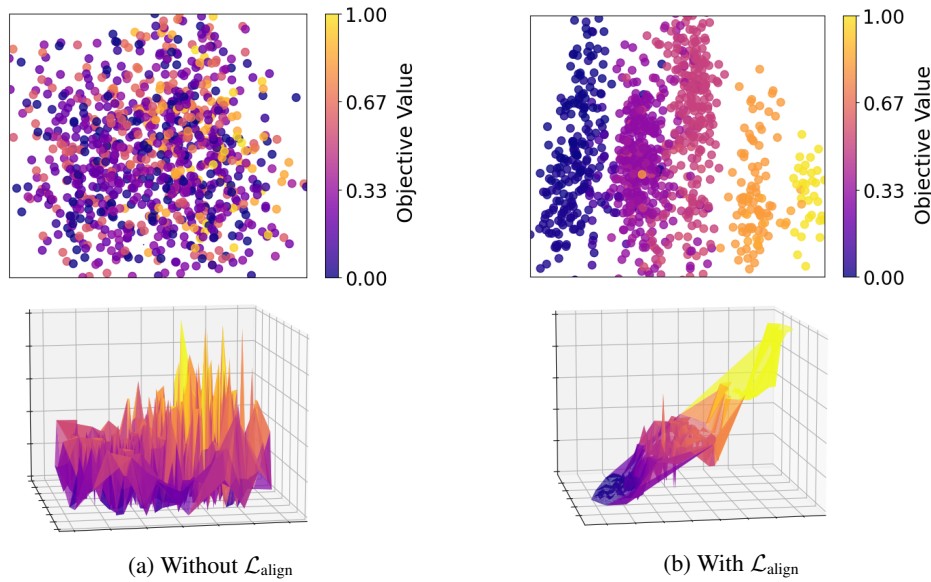

(a) Without $\mathcal{L}_{\text{align}}$        (b) With $\mathcal{L}_{\text{align}}$

Figure 5: **Visualizations on the landscape of latent space for $\mathcal{L}_{\text{align}}$ ablation.** The scatter plot and corresponding 3D plot of the latent vectors with objective values. The landscape becomes much smoother with applying $\mathcal{L}_{\text{align}}$. A colorbar indicates the normalized objective value, where yellow means higher value and purple means lower value.

## 4 Related Works

### 4.1 Latent space Bayesian optimization

Latent space Bayesian optimization [10, 12–14, 19, 21, 29, 30] aims to resolve the issues in optimization over high-dimensional, or structured input space by introducing Bayesian optimization over latent space. As the objective for structured inputs is usually defined over a large, complex space, the challenges can be alleviated by the lower-dimensional and continuous latent space. Variational autoencoders (VAEs) [15] are commonly leveraged to learn the continuous embeddings for the latent space Bayesian optimizers. Some prior works propose novel architectures for decoders [11, 12, 31–33], while others introduce loss functions to improve the surrogate for learning the objective function [13, 14, 18, 29]. Note that while the surrogate model (typically GP) is modeled based on the latent space, it is the input space which the objective value is obtained from. Although this results in an inherent gap in latent space Bayesian optimization, many prior works [12, 14, 19, 21, 29] do not update the generative model for the latent space. LOL-BO [10] seeks to address this gap by adapting the latent space to the GP prior, while [13] suggests periodic weighted retraining to update the latent space.

### 4.2 Latent space regularization

Several previous studies in latent space have focused on learning the appropriate latent space for their tasks by incorporating additional regularizations or constraints alongside the reconstruction loss. These approaches have been applied in a wide range of areas, including molecule design [34–36], domain adaptation [37, 38], semantic segmentation [39, 40], representation learning [41–43], and reinforcement learning [44]. Notably, the Lipschitz constraint is commonly employed to promote smoothness in diverse optimization problems. For example, [42] introduces Lipschitz regularization in learning implicit neural functions to encourage smooth latent space for 3D shapes while [45] penalizes the data pairs that violate the Lipschitz constraint in adversarial training. Additionally, CoFLO [46] leverages Lipschitz-like regularization in latent space Bayesian optimization. Our method proposes regularization to close the gap inherent in the latent space Bayesian optimization. Concretely, we introduce Lipschitz regularization to increase the correlation between the distance of latent space and the distance of objective value and give more weight to the loss in the promising areas.

# 5    Conclusion and Discussion

In this paper, we addressed the problem of the inherent gap in the latent space Bayesian optimization and proposed Correlated latent space Bayesian Optimization. We introduce Lipschitz regularization which maximizes the correlation between the distance of latent space and the distance of objective value to close the gap between latent space and objective value. Also, we reduced the gap between latent space and input space with a loss weighting scheme, especially in the promising areas. Additionally, by trust region recoordination, we adjust the trust regions according to the updated latent space. Our experiments on various benchmarks with molecule generation and arithmetic fitting tasks demonstrate that our CoBO significantly improves state-of-the-art methods in LBO.

**Limitations and broader impacts.** Given the contribution of this work to molecular design optimization, careful consideration should be given to its potential impacts on the generation of toxic or harmful substances in the design of new chemicals. We believe that our work primarily has the positive potential of accelerating the chemical and drug development process, setting a new standard in latent space Bayesian optimization.

## Acknowledgments and Disclosure of Funding

This work was partly supported by ICT Creative Consilience program (IITP-2023-2020-0-01819) supervised by the IITP, the National Research Foundation of Korea (NRF) grant funded by the Korea government (MSIT) (NRF-2023R1A2C2005373), and Samsung Research Funding & Incubation Center of Samsung Electronics under Project Number SRFC-IT1701-51.

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
