# Advancing Bayesian Optimization via Learning Correlated Latent Space (Supplement)

**Seunghun Lee**,* **Jaewon Chu**,* **Sihyeon Kim**,* **Juyeon Ko**,  **Hyunwoo J. Kim** [†]
Computer Science & Engineering
Korea University
{llsshh319, allonsy07, sh_bs15, juyon98, hyunwoojkim}@korea.ac.kr

**Summary.**   We provide additional experimental results/details and analysis in this supplement as: (A) analysis on regularization $\mathcal{L}_z$, (B) the proof of Theorem 1, (C) additional results on Guacamol Benchmarks, (D) additional results on DRD3 task, (E) efficiency analysis, and (F) implementation details.

## A   Analysis on Regularization $\mathcal{L}_z$

Here, we analyze the necessity of regularization $\mathcal{L}_z$. Based on Theorem 1 in the main paper, to increase the correlation between the distance of latent vectors and the differences in their corresponding objective values, we need to keep the distance between the latent vectors $\mathbf{z}$ to be a *constant*. Figure 1 displays the box plot of distances between $\mathbf{z}$ at each iteration of BO. The box represents the first and third quartiles, and the whiskers represent the 10 and 90 percentiles. Each data point has a top-$k$ score of objective value. As in Figure 1, the model only with Lipschitz regularization $\mathcal{L}_{\text{Lip}}$ (*i.e.*, without $\mathcal{L}_z$) increases the distance between the latent vectors $\|\mathbf{z}_i - \mathbf{z}_j\|_2$ since it is an easy way to minimize $\mathcal{L}_{\text{Lip}}$ given as

$$\mathcal{L}_{\text{Lip}} = \sum_{i,j \leq N} \max\left(0, \frac{|y_i - y_j|}{\|\mathbf{z}_i - \mathbf{z}_j\|_2} - L\right). \tag{1}$$

However, when applying both regularizations $\mathcal{L}_{\text{Lip}}$ and $\mathcal{L}_z$, we observe that the distance is preserved within a certain range, similar to the beginning of training.

## B   Proof of Theorem 1

**Theorem 1.** *Let $D_Z = d_Z(Z_1, Z_2)$ and $D_Y = d_Y(f(Z_1), f(Z_2))$ be random variables where $Z_1, Z_2$ are i.i.d. random variables, $f$ is an L-Lipschitz continuous function, and $d_Z, d_Y$ are distance functions. Then, the correlation between $D_Z$ and $D_Y$ is lower bounded as*

$$D_Y \leq LD_Z \Rightarrow \text{Corr}_{D_Z, D_Y} \geq \frac{\frac{1}{L}(\sigma_{D_Y}^2 + \mu_{D_Y}^2) - L\mu_{D_Z}^2}{\sqrt{\sigma_{D_Z}^2 \sigma_{D_Y}^2}},$$

*where $\mu_{D_Z}$, $\sigma_{D_Z}^2$, $\mu_{D_Y}$, and $\sigma_{D_Y}^2$ are the mean and variance of $D_Z$ and $D_Y$ respectively.*

---

*equal contributions
[†] Corresponding author

37th Conference on Neural Information Processing Systems (NeurIPS 2023).

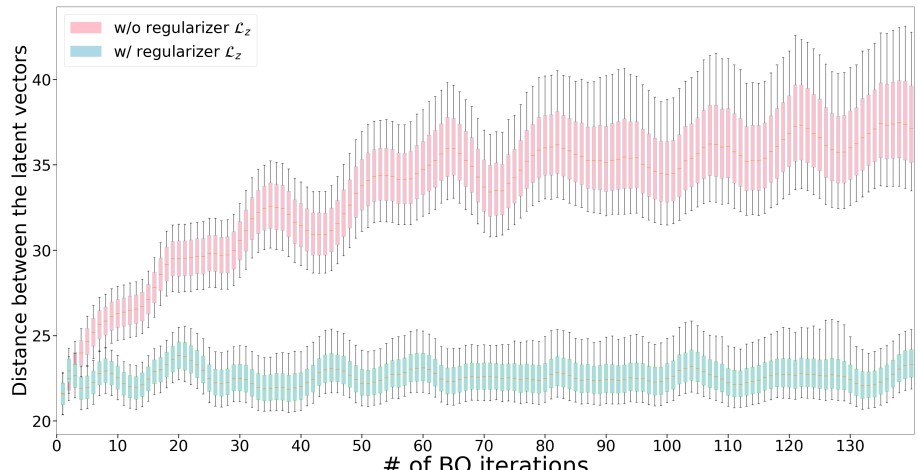

Figure 1: **Effects on Regularization $\mathcal{L}_z$.** The green and red box plots depict the distances of the latent vectors with and without regularization term $\mathcal{L}_z$, respectively.

*Proof.* The correlation between $D_Z$ and $D_Y$ is:

$$\text{Corr}_{D_Z, D_Y} = \frac{\text{Cov}\left(D_Z, D_Y\right)}{\sqrt{\text{Var}\left(D_Z\right)\text{Var}(D_Y)}} \tag{2}$$

$$= \frac{\mathbb{E}[(D_Z - \mathbb{E}[D_Z])(D_Y - \mathbb{E}[D_Y])]}{\sqrt{\text{Var}\left(D_Z\right)\text{Var}(D_Y)}} \tag{3}$$

$$= \frac{\mathbb{E}[D_Z D_Y] - \mathbb{E}[D_Z]\mathbb{E}[D_Y]}{\sqrt{\text{Var}\left(D_Z\right)\text{Var}(D_Y)}}. \tag{4}$$

By $L$-Lipschitz continuity, we have:

$$d_Y(f(Z_1), f(Z_2)) \leq L d_Z(Z_1, Z_2) \Rightarrow D_Y \leq L D_Z. \tag{5}$$

Hence, the correlation is bounded as follows:

$$\text{Corr}_{D_Z, D_Y} = \frac{\mathbb{E}[D_Z D_Y] - \mathbb{E}[D_Z]\mathbb{E}[D_Y]}{\sqrt{\text{Var}\left(D_Z\right)\text{Var}(D_Y)}} \tag{6}$$

$$\geq \frac{\mathbb{E}[\frac{1}{L}D_Y D_Y] - \mathbb{E}[D_Z]\mathbb{E}[L D_Z]}{\sqrt{\text{Var}\left(D_Z\right)\text{Var}(D_Y)}} \tag{7}$$

$$= \frac{\frac{1}{L}\mathbb{E}[(D_Y)^2] - L\mathbb{E}[D_Z]\mathbb{E}[D_Z]}{\sqrt{\text{Var}\left(D_Z\right)\text{Var}(D_Y)}} \tag{8}$$

$$= \frac{\frac{1}{L}\left(\text{Var}[D_Y] + (\mathbb{E}[D_Y])^2\right) - L(\mathbb{E}[D_Z])^2}{\sqrt{\text{Var}\left(D_Z\right)\text{Var}(D_Y)}} \tag{9}$$

$$= \frac{\frac{1}{L}(\sigma_{D_Y}^2 + \mu_{D_Y}^2) - L\mu_{D_Z}^2}{\sqrt{\sigma_{D_Z}^2 \sigma_{D_Y}^2}}. \tag{10}$$

$\square$

## C  Additional Results on Guacamol Benchmarks

In addition to the four tasks of the Guacamol benchmark that we previously mentioned, we also evaluate our model on three additional tasks: Ranolazine MPO, Aripiprazole similarity, and Valsartan SMART. The experimental settings for these additional tasks are the same with the settings applied

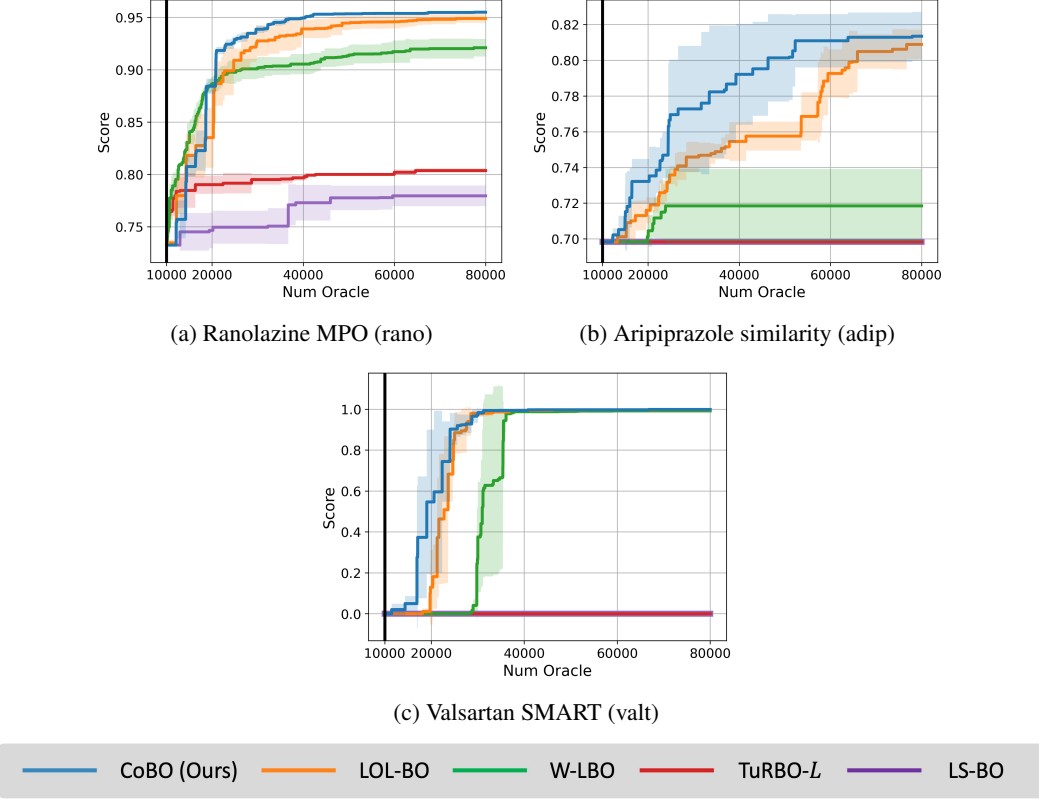

(a) Ranolazine MPO (rano)  (b) Aripiprazole similarity (adip)

(c) Valsartan SMART (valt)

| CoBO (Ours) | LOL-BO | W-LBO | TuRBO-$L$ | LS-BO |

Figure 2: **Optimization results with the additional three tasks on the Guacamol benchmark.** The lines and range are the mean and standard deviation of three repetitions with the same parameters.

to the initial four tasks. The results of the experiments are present in Figure 2. For the Valsartan SMART task, as depicted in Figure 2c, three models find the optimal point, note that our model finds the optimal point faster than other models.

## D  Additional Results on DRD3 Task

We compare our results with the leaderboard[3] of the DRD3 task in Table 1. Note that we use a random initialized dataset of 100. We specifically compare the Top-1 scores as absolute values, which are also reported in our line plot.

Table 1: Optimization results with best score on TDC DRD3 task. Baselines are reported on leaderboard.

| Oracle calls | CoBO (Ours) | Graph-GA[1] | SMILES-LSTM[2] | GCPN[3] | MARS[4] | MolDQN[5] |
|---|---|---|---|---|---|---|
| 100 | **-11.80** | -11.13 | -11.77 | -9.10 | -7.02 | -11.63 |
| 500 | **-13.57** | -12.50 | -11.37 | -11.97 | -9.83 | -7.62 |
| 1000 | **-13.97** | -13.23 | -11.97 | -12.03 | -11.10 | -7.80 |
| 3000 | **-15.37** | - | - | - | - | - |

## E  Efficiency Analysis

We conduct an efficiency analysis on every tasks: the Guacamol benchmarks, the DRD3 task, and the arithmetic fitting task. In our analysis, we compare our model with four baseline models. For a

---

[3]https://tdcommons.ai/benchmark/docking_group/drd3/

fair comparison, we set up experiments for every model in the same condition, as we use the CPU of AMD EPYC 7742 with a single NVIDIA RTX 2080 TI. Note that since these are CPU-intensive tasks, the CPU is crucial to the speed of execution. We report runtimes, the found best score, and the number of oracles, where we measure runtimes of executing a certain number of oracles as the wall clock time. The number of oracle calls increases each time as unique inputs are passed to the black-box objective function. Table 2, 3, 4 demonstrates that CoBO achieves comparable runtime with the same number of oracle calls, while outperforming the baselines by finding superior solutions.

## F  Implementation Details

In our implementation, we use PyTorch[4], BoTorch[5] and GPyTorch[6]. Additionally, we utilize the codebase[7] of [6] for the implementation. The SELFIES VAE is pretrained with 1.27M molecules in Guacamol benchmark and DRD3 task from [7] and the Grammar VAE is pretrained 40K expression in Arithmetic data from [8]. On the DRD3 task, we modify the evaluation metric from minimization to maximization by simply changing the sign of the objective values. In our experiments, we mainly employ NVIDIA V100 and Intel Xeon Gold 6230. In this setup, the pdop tasks with a budget of 70k oracle, took an average of 11 hours.

### F.1  Hyperparameters

We grid search coefficients of our proposed regularizations $\mathcal{L}_{\text{Lip\_W}}$ and $\mathcal{L}_{\text{z}}$, in the range of $[10,100,1000]$ for $\mathcal{L}_{\text{Lip\_W}}$ and $[0.1,1]$ for $\mathcal{L}_{\text{z}}$. For some tasks, we didn't search for these hyper-parameters, and their coefficients are provided in Table 6. The selected coefficients from this search are presented in Table 5. For other hyperparameters, such as coefficients for other losses, batch size, and learning rate, we set values according to Table 7.

---

Table 2: **Efficiency comparison on Guacamol benchmarks within 70k evaluation budget.**

| | Model | CoBO (Ours) | LOL-BO | W-LBO | TuRBO-*L* | LS-BO |
|---|---|---|---|---|---|---|
| med2 | Oracle calls | 70k | 70k | 70k | 70k | 70k |
| | Wall clock time (min) | 1057.8 | 1080.4 | **175.8** | 246.7 | 1580.3 |
| | Found Best Score | **0.3828** | 0.3530 | 0.3118 | 0.3118 | 0.3464 |
| | Oracle calls | 55k | 33k | **70k** | 48k | 16k |
| | Wall clock time (min) | 175.8 | 175.8 | 175.8 | 175.8 | 175.8 |
| | Found Best Score | **0.3828** | 0.3434 | 0.3118 | 0.3118 | 0.3295 |
| adip | Oracle calls | 70k | 70k | 70k | 70k | 70k |
| | Wall clock time (min) | 4986.3 | 3340.7 | **198.2** | 236.3 | 1320.8 |
| | Found Best Score | **0.8133** | 0.8086 | 0.6983 | 0.6983 | 0.7186 |
| | Oracle calls | 32k | 28k | **70k** | 58k | 13k |
| | Wall clock time (min) | 198.2 | 198.2 | 198.2 | 198.2 | 198.2 |
| | Found Best Score | **0.7921** | 0.7466 | 0.6983 | 0.6983 | 0.7186 |
| pdop | Oracle calls | 70k | 70k | 70k | 70k | 70k |
| | Wall clock time (min) | 1020.9 | 1920.4 | **168.4** | 268.1 | 840.6 |
| | Found Best Score | **0.8343** | 0.7959 | 0.5855 | 0.5736 | 0.6514 |
| | Oracle calls | 33k | 28k | **70k** | 38k | 18k |
| | Wall clock time (min) | 168.4 | 168.4 | 168.4 | 168.4 | 168.4 |
| | Found Best Score | **0.8343** | 0.7948 | 0.5855 | 0.5233 | 0.6312 |
| rano | Oracle calls | 70k | 70k | 70k | 70k | 70k |
| | Wall clock time (min) | 3940.5 | 2820.9 | 340.6 | **276.1** | 2320.6 |
| | Found Best Score | **0.9550** | 0.9468 | 0.8045 | 0.7766 | 0.9226 |
| | Oracle calls | 42k | 41k | 55k | **70k** | 21k |
| | Wall clock time (min) | 276.1 | 276.1 | 276.1 | 276.1 | 276.1 |
| | Found Best Score | **0.9486** | 0.9433 | 0.8045 | 0.7766 | 0.9166 |
| valt | Oracle calls | 70k | 70k | 70k | 70k | 70k |
| | Wall clock time (min) | 560.3 | 760.5 | 304.1 | **234.2** | 1940.4 |
| | Found Best Score | **0.9982** | **0.9982** | 4e-14 | 4e-33 | 0.9917 |
| | Oracle calls | 51k | 38k | 57k | **70k** | 23k |
| | Wall clock time (min) | 234.2 | 234.2 | 234.2 | 234.2 | 234.2 |
| | Found Best Score | **0.9982** | 0.9942 | 4.8532e-14 | 4875e-36 | 0.6533 |
| zale | Oracle calls | 70k | 70k | 70k | 70k | 70k |
| | Wall clock time (min) | 374.7 | 1320.2 | 366.7 | **150.4** | 840.5 |
| | Found Best Score | **0.7733** | 0.7521 | 0.6024 | 0.5142 | 0.6366 |
| | Oracle calls | 46k | 24k | 35k | **70k** | 11k |
| | Wall clock time (min) | 150.4 | 150.4 | 150.4 | 150.4 | 150.4 |
| | Found Best Score | **0.7733** | 0.7415 | 0.5633 | 0.5142 | 0.5833 |
| osmb | Oracle calls | 70k | 70k | 70k | 70k | 70k |
| | Wall clock time (min) | 477.6 | 840.1 | 372.2 | **210.2** | 743.4 |
| | Found Best Score | **0.9267** | 0.9233 | 0.8336 | 0.8481 | 0.8933 |
| | Oracle calls | 59k | 43k | 46k | **70k** | 19k |
| | Wall clock time (min) | 210.2 | 210.2 | 210.2 | 210.2 | 210.2 |
| | Found Best Score | **0.9233** | 0.9167 | 0.8332 | 0.8481 | 0.8866 |

Table 3: **Efficiency comparison on DRD3 benchmark within 3k evaluation budget.**

| Model | CoBO (Ours) | LOL-BO | W-LBO | TuRBO-$L$ | LS-BO |
|---|---|---|---|---|---|
| Oracle calls | 3k | 3k | 3k | 3k | 3k |
| Wall clock time (hr) | 86.4 | 65.7 | 40.7 | **34.3** | 67.4 |
| Found Best Score | **-15.4** | -14.6 | -12.3 | -12.2 | -13.9 |
| Oracle calls | 1.5k | 1.5k | 2.6k | **3k** | 2.1k |
| Wall clock time (hr) | 34.3 | 34.3 | 34.3 | 34.3 | 34.3 |
| Found Best Score | **-14.5** | -14.2 | -12.3 | -12.2 | -13.6 |

Table 4: **Efficiency comparison on arithmetic expression fitting task within 500 evaluation budget.**

| Model | CoBO (Ours) | LOL-BO | W-LBO | TuRBO-$L$ | LS-BO |
|---|---|---|---|---|---|
| Oracle calls | 500 | 500 | 500 | 500 | 500 |
| Wall clock time (min) | **5.4** | 11.3 | 11.1 | 338.1 | 1620.7 |
| Found Best Score | **0.1468** | 0.4624 | 0.5848 | 0.7725 | 0.5533 |
| Oracle calls | **500** | 330 | 173 | 0 | 0 |
| Wall clock time (min) | 5.4 | 5.4 | 5.4 | 5.4 | 5.4 |
| Found Best Score | **0.1468** | 0.5467 | 1.0241 | 1.521 | 1.521 |

Table 5: **Coefficients of our proposed regularizations determined by grid search.**

| | med2 | osmb | pdop | zale | Arithmetic | DRD3 |
|---|---|---|---|---|---|---|
| Coefficient of $\mathcal{L}_{\text{Lip\_W}}$ | 1e3 | 1e2 | 1e2 | 1e3 | 1e1 | 1e1 |
| Coefficient of $\mathcal{L}_{\text{z}}$ | 1e0 | 1e0 | 1e-1 | 1e0 | 1e-1 | 1e0 |

Table 6: **Coefficients of our proposed regularizations w/o search.**

| | rano | adip | valt |
|---|---|---|---|
| Coefficient of $\mathcal{L}_{\text{Lip\_W}}$ | 1e2 | 1e2 | 1e2 |
| Coefficient of $\mathcal{L}_{\text{z}}$ | 1e-1 | 1e-1 | 1e-1 |

Table 7: **Other hyperparameters used in the experiments.**

| Parameter | Guacamol | Arithmetic | DRD3 |
|---|---|---|---|
| Learning rate | 0.1 | 0.1 | 0.1 |
| Coefficient of $\mathcal{L}_{\text{surr}}$ | 1 | 1 | 1 |
| Coefficient of $\mathcal{L}_{\text{recon\_W}}$ | 1 | 1 | 1 |
| Coefficient of $\mathcal{L}_{\text{KL}}$ | 0.1 | 0.1 | 0.1 |
| Quantile of objective value for loss weighting | 0.95 | 0.95 | 0.95 |
| Standard deviation $\sigma$ for loss weighting | 0.1 | 0.1 | 0.1 |
| # initial datapoints $N$ | 10000 | 40000 | 100 |
| Latent update interval $N_{\text{fail}}$ | 10 | 10 | 10 |
| Batch size | 10 | 5 | 1 |
| # top-$k$ used training | 1000 | 10 | 10 |