# OpenReview forum: "Advancing Bayesian Optimization via Learning Correlated Latent Space"
_NeurIPS.cc/2023/Conference — NeurIPS 2023 poster_

### Official Review · Reviewer_ynAb · 2023-06-20

**Soundness:** 3 good
**Presentation:** 3 good
**Contribution:** 2 fair
**Rating:** 4
**Confidence:** 5

**Summary:**

Recent advances in Bayesian optimization have shown that it is possible to exploit latent spaces of variational auto-encoders or generative models to perform the optimization of any function defined over a structured space. However, since the optimization takes place in the latent space. There is an inherent gap between the original problem formulation and the optimization of the latent space. This paper proposed a series of new losses based on the smoothness map of the latent space to enforce a correlation between the function values in the latent space and the original space. The authors propose several losses based on Lipschitz constants and local searches to improve the optimization properties. Finally, the author proposes a series of experiments to show the benefit of their method.

**Strengths:**

I liked Theorem 1 which provides a nice theoretical justification of the innovations of the paper.

Moreover, it provides a nice storyline to explain the main elements of the paper and the loss introduced.

The paper is very clear and easy to follow.

There are a lot of numerical experiments showing the benefits of the method.

**Weaknesses:**

Overall, I really like the problem tackled in the paper. However, I have some questions with regards to the contributions, and more particularly with regards to lots of heuristics used in the algorihm:

- although there is a nice theoretical justification of the new losses introduced (Eq 3, 4 and 9), the method presented in Algorithm 1 relies on a large number of parameters such as k, the kernel used in the Gaussian process, the batch size, the latent space update interval Nfail which might make the method hard to use in practice. For instance, it seems like some parameters presented in the Appendix such as the batch size and k are different depending on the use case. Is there a way to know how to choose those hyperparameters that work in most cases?

- Similarly, for the halving of the trust region. It seems to be a full heuristic. Is there a justification for the value of the frequency to half the search space?

- In Eq 7, how to you choose the value of $y_q$? What is the precise meaning of this term in the loss?

- Bayesian optimization is mainly used when the black-box function is expensive (or costly) to evaluate. Moroever, there exists a large number of other methods for black-box optimization when the function is cheap to evaluate. However, when looking at the different experiments, it seems like the experiments can go up to 60K+ iterations. In this case, how would those methods compare to existing techniques (such as fine-tuned simulated annealing, genetic algorithms, and partitioning techniques such as the DIRECT algorithm) or even multi-start local search methods? Does using Bayesian optimization make a real difference?

- Overall and as a last comment, since the method relies on a large number of hyper-parameters, it seems like it is not realistic to choose them in practice in the case of expensive-to-evaluate black-box functions (which is the core justification of Bayesian optimization) in which functions evaluations are scarce. How could we improve that in practice?


**Questions:**

The questions raised have been asked above.

**Limitations:**

The limitations are quickly summarized in the conclusion. It could be nice to have a larger section in the Appendix

---

> ### Author Rebuttal · Authors · 2023-08-10
>
> We highly value the feedback provided by the reviewer. We offer responses to address the issues below.
>
> **Q1. Is there a justification for the value of the frequency to halve the search space?**
>
> We adopt every setting for the Trust region from TURBO [16], which is an established technique. For example, we use 32 for the number of consecutive failures as in [16]. We also followed [16], without specifically tuning the value of the frequency to halve the search space.
>
> **Q2. In Eq 7, how do you choose the value of $y_q$? What is the precise meaning of this term in the loss?**
>
> The value of $y_q$ corresponds to a specific quantile of the distribution of $y$. We use the value of $y_q$ as the same as in [21]. We assign weights based on Gaussian Cumulative Distribution Function (CDF), which is centered on $y_q$.
>
> **Q3. Compared to existing techniques like simulated annealing, genetic algorithms, or partitioning techniques, does using Bayesian optimization make a significant difference?**
>
> Here, our work aims to address the challenges of current (latent) Bayesian Optimization approaches. As a result, our primary focus for comparison is directed towards BO methods. While we acknowledge the potential for valuable insights through comparisons with conventional methods, we will add comparisons in the final version.
>
>
> **Q4. More detailed limitations**
>
> We will improve the limitations and reflect on Appendix in the final version.

---

### Official Review · Reviewer_aw6W · 2023-07-05

**Soundness:** 2 fair
**Presentation:** 2 fair
**Contribution:** 2 fair
**Rating:** 3
**Confidence:** 4

**Summary:**

The paper addresses the problem of Bayesian optimization with the help of a low-dimensional latent space (here learned using a VAE augmented with ad-hoc loss terms).

The paper contributes:
- An analysis/recap of the state-of-the-art highlighting issues raised in other works, particularly the need for smoothness (in the latent space) around the optima.
- A concrete loss function augmenting previous work [10] with two additional regularisation terms to ensure smoothness in the latent space w.r.t. the objective function.
- Experimental study demonstrating the advantages over a suitable set of benchmarks on six problems.
- An ablation study examining the effect of some of the loss terms on the final BO task.

**Strengths:**

-	Important and relevant research area.
-	The background and motivation for the derived algorithm are well-described.
-	The included experiments indicate improvements compared to a set of reasonable baselines.
-	The resulting algorithm seems sufficiently novel and builds on the intuition about smoothens w.r.t. to the objective function, but I have some concerns/questions…


**Weaknesses:**

Overall weaknesses/comments/suggestions:

-	The setup does not come across as principled or consistent from a pure modeling perspective…
  -	I appreciate the algorithmic-centered view (vs. a probabilistic modeling perspective) adopted in the paper. Still, I am always a bit skeptical when rather ad-hoc loss terms are added to the VAE loss or the joint setup from [10] (which both originate from a proper probabilistic analysis). The combination of and interaction among the many loss terms (in particular the $L_z$, $L_lip$, vs. the VAE prior) would need quite a lot of analysis to work out the exact effect of the combination (and weighting). Currently, I feel the paper lacks this insight.
 - $z$ is a random variable, yet the computation of the 2-norm in Eq. 9 is performed without recognizing this. I’d expect an expectation w.r.t.. q(z|x) to be involved in $L_z$ and $L_{lip_W}$ (or a non-central chi distribution if treated analytically)?
 -	I am unsure why $L_z$ is needed – can this not be controlled directly via the variance on the prior, $p(z)$? Can the smoothness constraints be incorporated/formulated as a more principled prior for z, thus providing a complete probabilistic view and derivation of the algorithm?
-	Missing assumptions.
 -	I think it needs to be argued that it makes sense to globally measure distance in the latent space using a Euclidian distance function.
 - It is generally unclear if you assume y is noiseless or not. This seems important in Eq. 9 (and Theorem 1), i.e., do you use $p(y|x, \cdot)$ or simply the observed y when computing $|y_i-y_j$?
-	The model/loss is not fully specified in the paper, as far as I can tell. Specifically,
 -	The weights on the loss terms are left out in Eq 11, giving the impression that there are no hyperparameters related to the loss itself, yet the supplementary clearly indicates that some manual fine-tuning is needed. The main paper should be transparent about this. How sensitive is the performance to the weights on the loss terms, and how should they be set in practice?
 -	l 155: $L_{joint}$ should be explained in more detail to provide a self-contained definition of the model/algorithm.
 -	There are no details in the paper or supplementary about the likelihood ($p(x|z)$); I’d suggest adding more information about this beyond a reference to the particular VAEs.
 -	There are no details about the structure and complexity of the problems or the VAE (e.g. dimensionality of z and its influence)$); I’d suggest providing this at least in the supplementary.
-	Experimental suggestions:
 -	If possible, I would suggest providing a very simple 2D (maybe 3D ) synthetic example (trained with the proposed model) to provide the reader with a better intuition about what’s going on with the many loss terms (c.f. previous comment).
 -	The ablation study (on one dataset) is interesting yet should probably include removing the $L_{joint}$ term as well.


Minor questions/comments:

-	I’d suggest providing (possibly in the appendix) an optimization trace for one example with the total loss and all the individual loss terms included individually.
-	l120: I think adaptively changing setting L using the median would require more justification and explanation to clarify the properties of this scheme.
-	l212: I am unsure what is meant with “….Thompson sampling in $N(0,I)$...”?
-	Figure 5: Is this a result based on a VAE with a latent space with dimension 2 or a projection onto 2D?
 - l 263: Fig 5 (b) seems less smooth than Fig 5 (a) as is; consider making the statement clearer (i.e. wrt to the objective)
-	Most figures lack indication of axis labels (on the figure itself or in the caption)
-	It would be helpful with a few more words/details attached to Eq 7 (the proof) in the supplementary material.
-	Some sections contain grammatical issues interrupting the flow, e.g. sec 2.4, l. 165  (missing “the”, “a” etc.)



**Questions:**

The questions are included in the above.

**Limitations:**

See above for technical limitations/issues. There is no need to consider the broader societal impact of the specific paper.

---

> ### Author Rebuttal · Authors · 2023-08-10
>
> We appreciate the reviewer's valuable feedback. We present comprehensive responses below.
>
> **Q1. Unsure why $\mathcal{L}\_z$ is needed**
>
> The $\mathcal{L}\_{\mathbf{z}}$ serves a specific purpose in our approach. If we were to rely solely on the Lipschitz loss and exclude $\mathcal{L}\_{\mathbf{z}}$, it could lead to a trivial solution where the scale of $\mathbf{z}$ is merely growing. This would be shown in the expression ${|y\_i-y\_j| \over ||\mathbf{z}\_i-\mathbf{z}\_j||\_2}$, which is the Lipschitz loss we minimize, resulting in a collapse of the distance metric in the denominator. To prevent this and ensure a meaningful scaling of $\mathbf{z}$, we introduced the $\mathcal{L}\_{\mathbf{z}}$ loss term. Furthermore, as described in lines 124-125, this term is closely related to Theorem 1, providing additional theoretical justification for its inclusion. We agree that the VAE prior can control these aspects, however, we expect the current formulation of $\mathcal{L}\_{\mathbf{z}}$ provide a targeted and effective solution to a specific challenge we identified in our approach. We also empirically verified its effectiveness.
>
> **Q2. I think it needs to be argued that it makes sense to globally measure distance in the latent space using a Euclidian distance function.**
>
> We appreciate your insight regarding the measurement of distance function. In the context of the Variational Autoencoder, the Euclidean space assumption is widely adopted [a], and this assumption guided our choice to measure distances in the latent space using the Euclidean distance function. It would be good for future work to define and use the geodesic distance on the manifold based on the Riemannian manifold [b].
>
> [a] Kalatzis, D., Eklund, D., Arvanitidis, G., & Hauberg, S. Variational Autoencoders with Riemannian Brownian Motion Priors. International Conference on Machine Learning, 2020.
> [b] Pennec, X. Intrinsic statistics on Riemannian manifolds: Basic tools for geometric measurements. Journal of Mathematical Imaging and Vision, 2006.
>
> **Q3. l 263: Fig 5 (b) seems less smooth than Fig 5 (a) as is; consider making the statement clearer.**
>
> We understand your point regarding Figure 5 (b) appearing less smooth than Figure 5 (a). It is possible to interpret the discontinuous portion in the middle of Figure 5 (b) as lacking smoothness. However, our intention was to illustrate the smoothness of the objective value at each location, rather than the visual continuity of the plot itself. Each color in the figure represents a different objective value, and we aimed to show that nearby points have similar colors, reflecting similar objective values. We will clarify the explanation of visual representation in Figure 5 (b).
>
> **Q4. l120: I think adaptively changing setting L using the median would require more justification and explanation to clarify the properties of this scheme.**
>
> We choose $L$ as the median to create a more robust and adaptive mechanism that can respond to varying properties within the data. This adaptive process allows the model to dynamically align with the characteristics of the data at different stages of training or across different datasets. This alleviates sensitiveness of the data and enhance stability in the learning process. We will provide additional analysis in the final version.
>
> **Q5. Unsure on “….Thompson sampling in $N(0,I)$...”?**
>
> We aim to convey is that in our approach using Gaussian Processes (GP), we sample the function and then draw candidate samples from the normal distribution $N(0,I)$ to find the optimal input value.
>
> **Q6. Do you use or simply the observed y when computing $|y\_i-y\_j|$?**
>
> In all the tasks we run, the objective function was noiseless, and we also assumed that y is noiseless. We will clarify this in the final version.
>
> **Q7. How does the combination and interaction of the various loss terms (such as $\mathcal{L}\_z$, $\mathcal{L}\_{lip}$, and the VAE prior) work, and what analysis has been done to understand their exact effect, considering the current paper seems to lack this insight?**
>
> The design of these loss terms each has a crucial role to achieve a specific goal including Lipschitz regularization and latent space regularization. Also, we showcased and analyzed that these loss terms lead to effective optimization in the latent space. We agree that further theoretical analysis would make the paper stronger and will add in the final version.
>
> **Q8. $\mathbf{z}$ is a random variable, yet the computation of the 2-norm in Eq. 9 is performed without recognizing this. I’d expect an expectation w.r.t.. q(z|x) to be involved in $\mathcal{L}_z$ and $\mathcal{L}\_{lip\\_W}$ (or a non-central chi distribution if treated analytically)?**
>
> Your expectation is correct. As you say, we were calculating the loss using the sampled z from Eq.9 and some other equations. We were using the bold to denote the vector. We will remove the bold for distinguishing sampled z and clarify it.
>
>
> **Q9. Additional questions/suggestions.**
>
> Due to limited space, we were unable to address all raised issues. We commit to correcting all typos and clarifying notations, adding analyses based on your valuable suggestions for the final version.

---

> > ### Comment · Reviewer_aw6W · 2023-08-15
> >
> > Thanks to the authors for responding to my (and the other reviewers') comments/questions.
> >
> > Re Q1 hyperparameters (summary): Could the authors please elaborate on the statement, “The grid search was each conducted for 20 percent of the total Oracle budget.” including an explanation of when/where exactly this was performed in the process?
> >
> > Re aw6W-Q1: Thanks for clarifying Eq. 4, and I now appreciate the desire for explicit control over the average distances relying not on $p(z)$. Although I suppose $L_z$ becomes redundant when the aggregated posterior and the prior are very similar, leading to the expected distances given by Eq 5…? I’d suggest providing strong arguments as to why the prior is insufficient to ensure desired properties in general.
> >
> > Re aw6W-Q7: Please clarify what “ …further theoretical analysis….” implies (i.e., what can we expect in a final version)?
> >
> > Overall, I remain slightly skeptical of the paper given the many updates needed/promised, but I am leaning towards increasing my score slightly.

---

> > > ### Author Response · Authors · 2023-08-18
> > >
> > > Thank you for your continued discussion. Here are our responses to the points raised:
> > >
> > > **Q1. More explanation of the hyperparameter search.**
> > >
> > > First, we tuned only two hyperparameters: the loss weights for $\mathcal{L}\_{\mathrm{lip}}$ and $\mathcal{L}_\mathbf{z}$. We performed a grid search with 6 combinations, i.e., [10, 100, 1000] x [0.1, 1] for the two losses. To pick the best hyperparameters, we preliminarily run experiments with 20% of the maximum query budgets. For instance, we set the maximum budget to 70K for Guacamol dataset and simply pick the hyperparameters with the best performance for 14K. So, the total query budget for grid search is 14K x 6 oracle calls.
> > >
> > > **Q2. Provide arguments as to why the prior is insufficient. (why $\mathcal{L}_\mathbf{z}$ needed)**
> > >
> > > We want to optimize a latent space allowing various distributions of latent $\mathbf{z}$ and we believe that the prior, KL divergence between the standard normal distribution and embeddings $\mathbf{z}$, is stricter than what we need. The prior explicitly requires 0 mean and Identity covariance. Also, the mode of a standard normal distribution is dense and many solutions around the mean have very small pairwise distances. On the other hand, $\mathcal{L}_\mathbf{z}$ does NOT require any specific mean, and the distribution of $\mathbf{z}$ does not need to be either isotropic or unimodal.  Lastly, our regularization encourages the distance between solutions in the latent space to be not either too small or large  $\| \mathbf{z}\_i - \mathbf{z}\_j \|_2 \approx c$.
> > >
> > > **Q3. Clarification on “further analysis regarding the combination and interaction of the various loss terms (VAE loss vs $\mathcal{L}\_\mathrm{lip}$ and $\mathcal{L}\_\mathbf{z}$)”.**
> > >
> > > - In Section 3.3 of our main paper, we have presented analyses concerning proposed regularizations. For instance, Figure 4 illustrates the higher correlation between the objective value $y$ and the latent vector $\mathbf{z}$ when employing $\mathcal{L}\_\mathrm{lip}$ and $\mathcal{L}\_\mathbf{z}$. Also, we show a smoother latent space with our proposed regularizations in Figure 5.
> > > - What "further analysis" meant was to conduct additional analyses of the *individual* impacts of each regularization term as you suggested. For example, to understand how does $\mathcal{L}\_\mathrm{lip}$, one of our regularizations, affects the landscape of the latent space, we conducted preliminary experiments with 2D synthetic dataset (generated samples with grid sampling) and trained a VAE model with and without $\mathcal{L}\_\mathrm{lip}$. We chose the Ackley function as the objective function since it is a non-convex function with a large number of local minima, i.e., it has a *non-smooth* landscape. After training, we drew the plots of the latent vectors with corresponding objective values and observed a more locally smooth landscape with $\mathcal{L}\_\mathrm{lip}$. Since we cannot provide more figures at this moment, we will incorporate these analyses into the Appendix of the final version.
> > > - Furthermore, we will include an additional smoothness analysis similar to that in Section 3.3. We will provide plots for two more cases: 1) $\mathcal{L}\_\mathrm{lip}$ alone and 2) VAE loss with $\mathcal{L}\_\mathrm{lip}$. Note that the results presented in Figure 5 of the main paper correspond to the cases of 1) only VAE loss and 2) VAE loss+$\mathcal{L}\_\mathrm{lip}$+$\mathcal{L}\_\mathbf{z}$.

---

### Official Review · Reviewer_4zSv · 2023-07-05

**Soundness:** 2 fair
**Presentation:** 3 good
**Contribution:** 2 fair
**Rating:** 4
**Confidence:** 5

**Summary:**

This paper proposes several heuristic regularization constraints for learning a Bayesian optimization latent space. It argues that the learned latent space needs to be aligned to the black-box function values, and this is achieved via keeping the Lipschitz constant small and the mean latent distance (of training samples) constant. It also argues that promising points need to be prioritized during the latent space optimization, and this is achieved via weighting the reconstruction loss and the Lipschitz regularization loss above at each training point by its function values. The paper evaluates the proposed method on several benchmark functions, yielding positive results.

**Strengths:**

The proposed approach makes some practical sense, and yields good performance on several benchmarks. The approach is also novel, so I believe the paper has some intellectual and practical merits.

**Weaknesses:**

1. I think the problem is not very clearly described and motivated. What exactly are the "gaps" between the latent space and the input space? Throughout the paper, I have not seen any technical description of this issue. To quote the manuscript, the two main gaps are:
- "First, even though the surrogate model g learned in the latent space the objective value is still obtained by a black-box function defined in the discrete input space X so the gap between objective values and latent spaces leads to poor optimization performance".
- "Second, since the distribution of samples expected to have high objective values is different from that of samples observed in pretraining, there is a gap between the input space and the latent space that makes the optimization inaccurate".

Neither statements give a precise description of what these gaps are, so I cannot be convinced that the proposed solutions are meaningful.

2. A lot of the technical details also lack motivation. Please refer to the specific questions below.

3. Different BO approaches have different per-iteration cost, and it would be slightly unfair to only compare the performance vs. number of oracle calls. I would strongly suggest providing a plot to show performance vs. wall-clock time, or a table documenting the runtime per iteration of each baseline.

**Questions:**

1. I don't understand why $\mu_{D_Z}$ and $\sigma_{D_Z}$ are not constants the same way $\mu_{D_Y}$ and $\sigma_{D_Y}$ are? And what are they constants with respect to?

2. Eq. (4) seems like a much stricter objective than what is needed -- it means that the points are optimized towards being equidistant. Also, when every $||z_i - z_j||_2 \simeq c$, what will remain for L_lip to optimize?

3. Why does blurring y with Gaussian noise in Eq. (7) make sense? Couldn't we just directly let $\lambda(y) = y_q - y$  (and maybe normalize this by y_max so that the weight is between 0/1) ?


**Limitations:**

The authors have discussed broader impacts & limitations. I don't see any potential negative social impact.

---

> ### Author Rebuttal · Authors · 2023-08-10
>
> We appreciate the reviewer's valuable feedback. Below, we present comprehensive responses addressing the questions raised by the reviewer.
>
> **Q1. Performance comparison w.r.t. wall-clock time.**
>
> We provide the results on Guacamol dataset with respective tasks with respect to wall-clock time as requested. We report the Found Best Score corresponding to the same wall-clock time,  determined by the fastest baseline model to reach the 70K oracle calls.
> | zale | CoBO (Ours) | LOL-BO | W-LBO | TuRBO | LS-BO |
> | --- | --- | --- | --- | --- | --- |
> | Found Best Score | **0.773** | 0.745 | 0.606 | 0.566 | 0.511 |
> | Oracle calls | 53K | 33K | 13K | 46K | **70K** |
> | Wall clock time (hr) | **2.6** | **2.6** | **2.6** | **2.6** | **2.6** |
> |  |  |  |  |  |  |
>
> | med2 | CoBO (Ours) | LOL-BO | W-LBO | TuRBO | LS-BO |
> | --- | --- | --- | --- | --- | --- |
> | Found Best Score | **0.379** | 0.352 | 0.330 | 0.312 | 0.312 |
> | Oracle calls | 58K | 35K | 20K | **70K** | 58K |
> | Wall clock time (hr) | **2.9** | **2.9** | **2.9** | **2.9** | **2.9** |
> |  |  |  |  |  |  |
>
> | osmb | CoBO (Ours) | LOL-BO | W-LBO | TuRBO | LS-BO |
> | --- | --- | --- | --- | --- | --- |
> | Found Best Score | **0.928** | 0.910 | 0.898 | 0.834 | 0.844 |
> | Oracle calls | 67K | 43K | 21K | **70K** | 64K |
> | Wall clock time (hr) | **3.5** | **3.5** | **3.5** | **3.5** | **3.5** |
> |  |  |  |  |  |  |
>
> | pdop | CoBO (Ours) | LOL-BO | W-LBO | TuRBO | LS-BO |
> | --- | --- | --- | --- | --- | --- |
> | Found Best Score | **0.834** | 0.796 | 0.635 | 0.587 | 0.565 |
> | Oracle calls | 41K | 35K | 17K | **70K** | 52K |
> | Wall clock time (hr) | **2.7** | **2.7** | **2.7** | **2.7** | **2.7** |
> |  |  |  |  |  |  |
>
> **Q2. Why use Gaussian noise in Eq. (7) for blurring y instead of directly using $\lambda(y)=y_q-y$?**
>
> We opted for the weight function from [21], as it demonstrated superior performance in contrast to other weight functions. We performed preliminary experiments for weight function selection on Guacamol (w/ pdop). We provide the experimental results and formulations of each weight function in the PDF file. Among these weight functions, Rwr closely aligns with your suggestion (with the addition of a temperature parameter). It is clear that [21] (indicated by the blue line) showcases the most effective performance.
>
> **Q3. Why are $\mu_{D_Z}$ and $\sigma_{D_Z}$ are not constants the same way $\mu_{D_Y}$ and $\sigma_{D_Y}$ are? And what are they constants with respect to?**
>
> Since the input data points are associated with fixed objective values of the black box function, we can consider the $\mu_{D_Y}$ and $\sigma_{D_Y}$ as constants. However, the distance in the latent space (i.e., ${D_Z}$) depends on the mapping function (encoder). Because the mapping function can be changed during the optimization process, $\mu_{D_Z}$  and $\sigma_{D_Z}$  are not treated as constants.

---

> > ### Comment · Reviewer_4zSv · 2023-08-15
> > **Thank you for your response**
> >
> > I appreciate the new results. It also seems like you missed one of my question. Can you provide a discussion regarding my Q2?

---

> > > ### Author Response · Authors · 2023-08-15
> > >
> > > Thank you for your reply! Regarding Q2 (the second question in the "Question" section: about Eq. 4), we have already addressed this inquiry in the above section of our global response.  We kindly request you to review our response.

---

> > > > ### Comment · Reviewer_4zSv · 2023-08-18
> > > > **Discussion**
> > > >
> > > > Thanks for the clarification. I missed that from the global response. It does make a bit more sense now, but I still don't get it why should the average distance be optimized towards a specific constant c that is related to the prior?
> > > >
> > > > This is just one part of a bigger weakness, that the central problem of this paper is not well described and motivated. Throughout the paper, I can't pick out any technical description of the challenges. I hope to see other reviewers' takes on this, but as things are I am inclined to keep my original score.

---

> > > > > ### Author Response · Authors · 2023-08-21
> > > > >
> > > > > Thank you for your comments. I hope the answer below has clarified your understanding of the motivation of our approach.
> > > > >
> > > > > **Q1. Why is the average distance optimized to a specific constant $c$ related to the prior?**
> > > > >
> > > > > The optimization of average distance towards a constant $c$ prevents a trivial solution where the scale of z merely grows. The inclusion of the $\mathcal{L}_\mathbf{z}$ loss term and aligning it with a constant $c$ follows our Theorem 1. We theoretically derived $c$ respecting the prior of VAE. This removes the necessity of hyperparameter tuning for $c$.
> > > > >
> > > > > **Q2. What exactly is the central problem (gaps) in the paper?**
> > > > >
> > > > > The proposed method addresses two types of gaps.
> > > > >
> > > > > **1. Gap between objective function $\mathcal{f}$ and latent space $\mathcal{Z}$:**
> > > > >
> > > > > In Latent Space Bayesian Optimization, the objective function is defined in the input space $\mathcal{X}$, while the surrogate model is defined in the latent space $\mathcal{Z}$. But pretrained latent space does not consider the objective function, which is the black-box function. This misalignment between latent space and objective function can reduce the optimization performance when mapping objective values from the latent space. We address this by **“Lipschitz regularization”** and **“L_z regularization”**.
> > > > >
> > > > > **2. Gap between input space $\mathcal{X}$ and latent space $\mathcal{Z}$:**
> > > > >
> > > > > This gap is induced by the discrepancy between training samples and promising samples with high objective values. The latent space $\mathcal{Z}$, defined during pretraining, may not be optimal to represent promising samples with a high objective value within a trust region. We address this by **“loss reweighting”** and **“recoordination”**.

---

### Official Review · Reviewer_Bvzj · 2023-07-07

**Soundness:** 4 excellent
**Presentation:** 3 good
**Contribution:** 3 good
**Rating:** 7
**Confidence:** 4

**Summary:**

This paper proposes a latent space Bayesian Optimization approach based on the intuition that distances in the latent space should be correlated with differences in objective value. CoBO iteratively updates a variational autoencoder (VAE) to align distances in latent space with differences in objective function. The method leverages a lower bound on correlation to introduce new regularizers on the VAE objective. Combined with loss weighting and latent space “recoordination”, the method achieves state-of-the-art performance on diverse benchmarks.

**Strengths:**

The paper mostly applies existing techniques to achieve incremental but statistically significant improvement over prior state-of-the-art. Theorem 1 provides a useful result that is a significant contribution of its own, if it is in fact a new bound.

The paper is written very well and will likely serve as a benchmark for future BO algorithms. Algorithm 1 provides a nice, approachable, and thorough description of CoBO. Figures 4-5 show strong evidence that the CoBO objective has a strong positive effect on organizing the latent space.

The experimental results are convincing. An ablation study justifies the inclusion of all components of the proposed VAE objective, albeit on only one optimization benchmark (hopefully these results hold for all benchmarks studied).

I would suggest trying to fit Table 3 from the Supplemental into the body of the paper, as it would strengthen the experimental evaluation. The results show strong performance especially in the low-budget setting.

**Weaknesses:**

It does not seem natural to assume that the mapping $f$ is Lipschitz for a VAE trained in an unsupervised manner. Won’t it be the case that the objective is not Lipschits in the input space of many useful models, especially over discrete spaces? Specifically, do you have any experiments demonstrating Lipschitz continuity of the objective by sampling points in the latent space?

_Note that the “recoordination” technique claimed by this paper previously existed in Maus et al. under the name “recentering”._

Evaluation is a little limited. Do the results hold for all Guacamole tasks? For example, Maus et al evaluates on additional GuacaMol tasks such as ranolazine.

Evalutions on DRD3 only go up to 1,000 evaluations of the objective function. This is understandable given the expense of running the docking objective, but on the other hand, this is the most “real-world” objective studied. Would it be possible to run for longer to determine whether CoBO can eventually beat LOL-BO or GraphGA?

The axes for all plots start at 0; it would be more helpful to include initialization points in the evaluation budget for comparison with prior work.

Specific issues:
* Articles are missing in several locations, including in the abstract (line 12). Please proofread for small errors.
* Equation 4 shows a regularizer that encourages __every__ pair of distances to be exactly $c$; it does not regularize the mean as claimed in the text. Is the text or the equation incorrect?
* Do you recoordinate all points during fine tuning? If so, add this step to Algorithm 1 (I believe it currently only shows recentering for the trust region, not the surrogate).
* Is there a reason that you report the objectives in Arithmetic expressions as a minization problem, but transform the DRD3 task into a maximization task? Retaining the convention of the TDC leaderboard would improve the accessibility of the paper to a wide audience.

Small comments:
* Define “an inherent gap” in the abstract. This notion is referred to multiple times in the paper, but never precisely. Is the claim that there do not exist points in the latent space $z$ that correspond to an optimal point in input space $x$? Or that these regions are small and hard to find?
* It is not immediately obvious in the introduction that this method will leverage a pre-trained VAE that is trained in a purely unsupervised manner.
* In Equation 7, notation seems to be overloaded. Do you mean a quintile of the empirical distribution of objective values seen during training, $y_i$? At line 140, is $sigma$ a hyperparameter or equal to $\sigma_Y$?
* Algorithm 1 line 14: typo “is the best score than”
* When discussing “initialization points” in Section 3, it would be useful to the the notation $D_0$ from Algorithm 1.
* In Figure 2, should “TDL” be “TDC”?
* It would be helpful to label the y-axes in Figure 4.

**Questions:**

1. The paper ignores the fact that the VAE decoder in many applications is stochastic (e.g. the SMILES decoder from Maus et al.). This would motivate a probabilistic treatment of f(Z) in Theorem 1.
2. In Theorem 1, won’t it always be the case that $D_Y \le D_Z$ by the Lipschitz assumption? Why is this stated as a sufficient condition?
3. On line 120, is this Lipschitz constant computed as the median __per mini-batch__?
4. Equations 4-6 do not appear to be consistent. You derive the distribution of inter-point distances, so why regularize all distances to be the same? Do you think of this as an additional discrepancy measure between your VAE prior and posterior? Can this be connected to the MMD regularization in InfoVAEs?
5. The paper describes the method in terms of standard Gaussian Process kernels. Do you not use a deep kernel, even for the drug discovery tasks?
6. How are hyperparameters determined, particularly in the ablation study?
7. How do you choose the initialization points?
8. Does Figure 4 include the recoordination step?
9. In Figures 4-5, would it be possible to compare to the prior state-of-the-art (LOL-BO) to substantiate the claim that that method does not effectively reorganize the latent space, rather than simply comparing to a fixed latent space?

**Limitations:**

I believe that CoBO can only be expected to be effective on objectives that are Lipschitz continuous; this limitation is not mentioned. The author’s consideration of possible malicious uses for drug discovery models is much appreciated.

---

> ### Author Rebuttal · Authors · 2023-08-10
>
> We appreciate the thorough and detailed feedback. We will address the issues below.
>
> **Q1. More evaluations on DRD3.**
>
> We conducted the experiment with DRD3 for a duration of up to 5 days, reaching a maximum of 2500 oracle calls within the available resources. We also provide the optimization curve in the PDF file. CoBO shows the superior performance over baselines.
>
> | DRD3 | CoBO (Ours) | LOL-BO | W-LBO | TuRBO | LS-BO |
> | --- | --- | --- | --- | --- | --- |
> | Found Best Score | **15.6** | 14.0 | 14.2 | 12.4 | 12.2 |
> | Oracle calls | **2500** | **2500** | **2500** | **2500** | **2500** |
> |  |  |  |  |  |  |
>
> **Q2. Additional tasks for Guacamol.**
>
> We report three additional tasks (rano, valt, adip) for Guacamol dataset. With limited resources, we set the Oracle budget to 30K. To ensure fairness, we also include the Found Best Score for the same wall-clock time. We also provide the optimization curve in the PDF file. It demonstrated superior performance in additional tasks, aligning with the results of the main paper.
>
> | rano | CoBO (Ours) | LOL-BO | W-LBO | TuRBO | LS-BO |
> | --- | --- | --- | --- | --- | --- |
> | Found Best Score | **0.9548** | 0.9244 | 0.9155 | 0.8184 | 0.7775 |
> | Oracle calls | **50K** | **50K** | **50K** | **50K** | **50K** |
> | Wall clock time (hr) | 14.5 | 11.8 | 9.1 | 5.8 | **4.8** |
> |  |  |  |  |  |  |
> | Found Best Score | **0.9530** | 0.8329 | 0.9014 | 0.8184 | 0.7775 |
> | Oracle calls | 32K | 15K | 17K | 40K | **50K** |
> | Wall clock time (hr) | **4.8** | **4.8** | **4.8** | **4.8** | **4.8** |
> |  |  |  |  |  |  |
>
> | valt | CoBO (Ours)  | LOL-BO | W-LBO | TuRBO | LS-BO |
> | --- | --- | --- | --- | --- | --- |
> | Found Best Score | **0.9981** | 0.9961 | 0.9727 | 5e-15 | 2e-31 |
> | Oracle calls | **50K** | **50K** | **50K** | **50K** | **50K** |
> | Wall clock time (hr) | **3.2** | 10.9 | 21 | 11.5 | 8.0 |
> |  |  |  |  |  |  |
> | Found Best Score | **0.9981** | 0.9759 | 2e-26 | 2e-26 | 2e-31 |
> | Oracle calls | **50K** | 25K | 10K | 15K | 20K |
> | Wall clock time (hr) | **3.2** | **3.2** | **3.2** | **3.2** | **3.2** |
> |  |  |  |  |  |  |
>
> | adip | CoBO (Ours)  | LOL-BO | W-LBO | TuRBO | LS-BO |
> | --- | --- | --- | --- | --- | --- |
> | Found Best Score | **0.8321** | 0.7746 | 0.7817 | 0.6983 | 0.6983 |
> | Oracle calls | **50K** | **50K** | **50K** | **50K** | **50K** |
> | Wall clock time (hr) | 27 | 6.7 | 32.6 | **3.1** | **3.1** |
> |  |  |  |  |  |  |
> | Found Best Score | **0.8321** | 0.7518 | 0.7534 | 0.6983 | 0.6983 |
> | Oracle calls | 27K | 33K | 14K | **50K** | **50K** |
> | Wall clock time (hr) | **3.1** | **3.1** | **3.1** | **3.1** | **3.1** |
> |  |  |  |  |  |  |
>
> **Q3. Is it natural to assume that the mapping f is Lipschitz for a VAE pretrained unsupervisedly, especially over discrete spaces?**
>
> Good question! We do NOT assume that the pretrained VAE is Lipschitz continuous. That’s why we proposed the regularizers in Eq. 3 and 4 in the main paper. Also, we observe that although the input/output of VAE are discrete values, except for the first and last layers, the VAE internally treats the data like continuous variables. All intermediate features including latent variable $z$ are continuous values. Therefore, imposing a Lipschitz constraint on VAE is a reasonable remedy to improve the latent space and reduce the gap between the latent space and the unknown objective function.
>
> **Q4. Is the claim that there do not exist points in the latent space that correspond to an optimal point in input space? Or that these regions are small and hard to find?**
>
> Great question! In general, the latent space has a point that corresponds to an optimal solution in the input space. First, the VAE returns a probability mass function with p in (0,1). Since it cannot be strictly 0, or 1, the generated PMF covers all possible discrete solutions. In addition, the Gaussian distribution in the continuous latent space has a infinite support so the latent vector that corresponds to an optimal solution has non-zero density. Indeed, multiple latent vectors correspond to the same optimal solution. In sum, the optimal point must exist in the latent space but the regions can be small or the likelihood to get the optimal latent vector could be small.
>
> **Q5. The stochastic nature of the VAE decoder.**
>
> We treat the decoder as deterministic by considering the expectation $\mathbb{E}_\theta(p\_\theta(\mathbf{x}|\mathbf{z}))$. We acknowledge that considering the stochastic nature of the decoder can provide a more precise understanding, thus we will reflect to our future works.
>
> **Q6. About "recoordination".**
>
> The concept of both [10] and CoBO is similar, but there are some differences in implementation, especially in deciding the center. In [10], there's a possibility of retrieving the latent space before updating VAE, due to implementation issues. [10] determines the center based on latent space $\mathbf{z}$ with the decoded input w.r.t. the best objective value $y$, whereas ours uses the latent space $\mathbf{z}$ with the input w.r.t. the best objective value $y$. We empirically showed better performance over [10]. Also, we only recoordinate the center of the trust region, rather than all points during fine-tuning.
>
> **Q7. Do you have experiments showing Lipschitz continuity of the objective?**
>
> We agree that understanding the Lipschitz continuity of the objective value is an essential aspect of our study. To demonstrate the Lipschitz continuity, we measure the change of Lipschitz loss over multiple methods (i.e., we do not utilize Lipschitz loss for baselines). The curve is in the PDF file. The Lipschitz loss increased at the beginning of the training process, due to the differences in the observed objective values.
>
> **Q8. Additional questions/suggestions.**
>
> Due to limited space, we were unable to address all raised issues. We commit to correcting all typos and clarifying notations in line with your valuable suggestions, for the final version.

---

> > ### Comment · Reviewer_Bvzj · 2023-08-13
> > **Response to rebuttal**
> >
> > Thank you to the authors for a thoughtful rebuttal, and especially for providing new experiential results. I remain supportive of this paper, as it appears to be a current leading Bayesian optimization approach for de novo molecular design tasks. I remain conflicted, though, because of the limited methodological novelty and incremental improvement in results.
> >
> > After reviewing the additional experiments, I would like to verify that the authors are treating the DRD3 task as a _minimization_ problem. If the objective values shown in the plot are the _negative_ of the DRD3 objective on the [leaderboard](https://tdcommons.ai/benchmark/docking_group/drd3/), then it appears from the results in the rebuttal PDF that LOL-BO is performing better for small query budgets than reported in Maus et al.
> >
> > It appears that the new experiments are similar to the results in the paper: CoBO provides small gains over LOL-BO (e.g. for Ranolazine), especially for small query budgets.
> >
> > Can you confirm whether Theorem 1 provides a novel contribution or whether this is restating a well-known theory, in your view?
> >
> > Please ensure that you explain the difference between recentering and coordination in your camera-ready version to avoid confusion.
> >
> > If possible, please correct figure x-axes to include initialization points in the query budget.

---

> > > ### Author Response · Authors · 2023-08-15
> > >
> > > Thank you for your swift feedback. Here's a summary of our response:
> > >
> > > **Q1. Clarification on DRD3 task.**
> > >
> > > A1. In the Fig. 3 (b), we took out the minus sign from the function value and plotted it as a maximization problem. In the final version, we will ensure that the plot for the DRD3 task accurately represents it as a minimization problem as suggested. The performance difference between our experimental results in the attached PDF file and the original paper (Maus et al.) may come from hyperparameters. Since we couldn’t find the original implementation details for LOL-BO+DRD3 task, we independently searched for appropriate hyperparameters. Also, note that we could only conduct a single experiment for each model due to the limited time/resources. We will conduct more comprehensive experiments for the final version possibly with error bars.
> > >
> > > **Q2. Small gain over LOL-BO in additional experiments (for small query budgets).**
> > >
> > > A2. In the PDF file, the first figure on the top shows that CoBO outperforms every baseline in all four tasks, and even for small query budgets (e.g., < 20,000 oracle calls), CoBO shows superior performance (e.g., Adip, Rano(lazine)). We notice for Valt, CoBO may show a relatively small gain over LOL-BO when the performance is saturated.
> > >
> > > **Q3. Novelty on Theorem 1.**
> > >
> > > A3. To the best of our knowledge, our Theorem 1 is novel. We could not find a theory similar to Theorem 1.
> > >
> > > **Q4. Clarification for recentering/recoordination + Correction on figure (x-axes).**
> > >
> > > A4. Thank you for your thorough and detailed comments! We will clarify the ”recoordination” part in the final manuscript and correct the x-axes to include the initialization points.

---

> > > > ### Comment · Reviewer_Bvzj · 2023-08-15
> > > > **Thank you for the detailed answers**
> > > >
> > > > Thank you for the detailed answers. Primarily based on the promising DRD3 results, but also based on the apparent methodological novelty of Theorem 1, I am going to raise my score to a 7.
> > > >
> > > > I would encourage the authors to try to run multiple replicates of the DRD3 experiment to evaluate potential competitiveness with GraphGA in the high query regime. It would likely be worthwhile to submit this method to the TD Commons leaderboard.

---

> > > > > ### Author Response · Authors · 2023-08-15
> > > > >
> > > > > Thank you for your feedback and encouraging response! We will add comprehensive experiments for DRD3 task with multiple experiments in the final version as you suggested and look into submitting the performance on the leaderboard.

---

### Author Rebuttal · Authors · 2023-08-10

We thank all reviewers for their thorough and thoughtful feedback. We will address all issues raised by the reviewers below. Especially, we will provide answers to common questions that we have received from multiple reviewers in this General section.

**Q1. Regarding hyperparameter search and selection. (Bvzj, aw6W, ynAb)**

We grid search coefficients of our proposed regularizations $\mathcal{L}\_{\text{lip\\_w}}$ and $\mathcal{L}\_{\mathbf{z}}$, in the range of [10,100,1000]/[0.1,1]. The grid search was each conducted for 20 percent of the total oracle budget. The final coefficients are mentioned in Appendix E.1.

For other hyperparameters such as coefficients for other losses, batch size, learning rate, … etc, we fix the coefficient as in Table 5 from Appendix E.1, as we did not do further searching. Rather, we utilized hyperparameters from previous methods, e.g., the hyperparameters related to trust region in TURBO [16]. For the batch size, we chose values within the range allowed by GPU memory, and for initialization points, we mostly followed [15] and if exist, we followed the official evaluation settings from the benchmark dataset (e.g., DRD3).

We thank all reviewers for suggestions and we will reflect the above details in the Appendix for the final version.

**Q2. Confusion regarding Eq. 4 in the main paper. (Bvzj, 4zSv)**

We apologize for the confusion. Our intention is to control the average of the $\mathbf{z}$ distances, not to optimize for all distances between points to be equidistant. Thus, Eq. 4 should be revised as below:


$\displaystyle\mathcal{L}_{\text{z}} = {1\over N^2} \sum\_{i,j\le N} \bigg|||\mathbf{z}_i - \mathbf{z}_j||_2-c\bigg| \rightarrow\mathcal{L}\_{\text{z}} = \bigg|{1\over N^2} \sum\_{i,j\le N} ||\mathbf{z}_i - \mathbf{z}_j||_2-c\bigg|$

**We provide additional experiments on the attached PDF file.**
These are details on references in the PDF file:
[A] Rubinstein, R. The cross-entropy method for combinatorial and continuous optimization. Methodology and computing in applied probability, 1999.
[B] Tripp, A., Daxberger, E., & Hernández-Lobato, J. M. Sample-efficient optimization in the latent space of deep generative models via weighted retraining. Advances in Neural Information Processing Systems, 2020.
[C] Peters, J., & Schaal, S. Reinforcement learning by reward-weighted regression for operational space control. International conference on Machine learning, 2007.

---

### Decision · Program_Chairs · 2023-09-21

**Decision:**

Accept (poster)

**Comment:**

I think this is a pretty good paper, and I'm unconvinced that several concerns raised by reviewers are actually concerns specific to this paper. For example, Reviewer ynAb points out the modeling design choices that must be made, but the specific choices listed are going to be true for any TuRBO-like BayesOpt algorithm (e.g., choosing a trust region length) or some for even any BO algorithm in general (for example, the kernel). I don't think we should suddenly start penalizing this paper for the existence of these considerations, some of which have existed in the BayesOpt literature at least since it has used Gaussian processes. Reviewer aw6W argues that the paper might not be "principled from a modelling perspective," but I'm just not sure whether this should matter specifically on optimization tasks like those over molecules where guarantees seem essentially impossible to achieve. LOL-BO can be viewed as a joint variational inference problem, yes, but it's unclear if that actually helps us guarantee anything about optimization performance.

I do think there are a few concerns raised that the authors *should* address, beyond the extensive results they've included in response to Reviewer Bzvj:

- The authors are pretty fast and loose with terminology like the "inherent gap" between the objective function and the latent space. I know what the authors mean, but they should be more precise about this. In Maus et al., 2022 for example, it is described as correlations in the objective function not being well measured by the GP prior used to model the objective.

- I'm additionally unsure about the results on expressions and DRD3 -- Maus et al., 2022 reports substantially better performance on both tasks than seen in this paper (~0.1 on Expressions by 500 evaluations, and ~13.8+/-0.5 on DRD3 after 1000 evaluations). I'm curious about the source of this gap, as the results on the Guacamol tasks match fairly closely those in the original paper. Perhaps some hyperparameter (e.g., number of failures before shrinking the trust region) was tuned differently on those tasks due to the much smaller evaluation budgets involved?

Despite the above (and while I'd urge the authors to address them), I'm recommending the paper be accepted. Getting optimization working on these tasks is not easy. Even if the DRD3 and expressions results are fixed and much closer to matching at those evaluation budgets, I think the results are pretty impressive on the Guacamol tasks, and it seems as though the higher budget DRD3 results reported in the author feedback period will be better anyways.